# Magnetism-induced topological transition in EuAs₃

Erjian Cheng[1,15], Wei Xia[2,3,15], Xianbiao Shi[4,5,15], Hongwei Fang[2,6,15], Chengwei Wang[2,7], Chuanying Xi[8], Shaowen Xu[9], Darren C. Peets[10,11], Linshu Wang[1], Hao Su[2], Li Pi[8], Wei Ren[9], Xia Wang[2], Na Yu[2], Yulin Chen[2,3,12], Weiwei Zhao[4,5], Zhongkai Liu[2,3✉], Yanfeng Guo[2✉] & Shiyan Li[1,13,14✉]

The nature of the interaction between magnetism and topology in magnetic topological semimetals remains mysterious, but may be expected to lead to a variety of novel physics. We systematically studied the magnetic semimetal EuAs₃, demonstrating a magnetism-induced topological transition from a topological nodal-line semimetal in the paramagnetic or the spin-polarized state to a topological massive Dirac metal in the antiferromagnetic ground state at low temperature. The topological nature in the antiferromagnetic state and the spin-polarized state has been verified by electrical transport measurements. An unsaturated and extremely large magnetoresistance of ~2 × 10⁵% at 1.8 K and 28.3 T is observed. In the paramagnetic states, the topological nodal-line structure at the $Y$ point is proven by angle-resolved photoemission spectroscopy. Moreover, a temperature-induced Lifshitz transition accompanied by the emergence of a new band below 3 K is revealed. These results indicate that magnetic EuAs₃ provides a rich platform to explore exotic physics arising from the interaction of magnetism with topology.

[1] State Key Laboratory of Surface Physics, Department of Physics, and Laboratory of Advanced Materials, Fudan University, 200433 Shanghai, China. [2] School of Physical Science and Technology, ShanghaiTech University, 200031 Shanghai, China. [3] ShanghaiTech Laboratory for Topological Physics, 201210 Shanghai, China. [4] State Key Laboratory of Advanced Welding & Joining and Flexible Printed Electronics Technology Center, Harbin Institute of Technology, 518055 Shenzhen, China. [5] Key Laboratory of Micro-systems and Micro-structures Manufacturing of Ministry of Education, Harbin Institute of Technology, 150001 Harbin, China. [6] University of Chinese Academy of Sciences, 100049 Beijing, China. [7] State Key Laboratory of Functional Material for Informatics, Shanghai Institute of Microsystem and Information Technology, Chinese Academy of Sciences, 200050 Shanghai, China. [8] Anhui Province Key Laboratory of Condensed Matter Physics at Extreme Conditions, High Magnetic Field Laboratory of the Chinese Academy of Sciences, 230031 Hefei, Anhui, China. [9] Department of Physics, Shanghai University, 200444 Shanghai, China. [10] Ningbo Institute of Materials Technology and Engineering, Chinese Academy of Sciences, 315201 Ningbo, Zhejiang, China. [11] Institute for Solid State and Materials Physics, Technical University of Dresden, 01062 Dresden, Germany. [12] Department of Physics, University of Oxford, Oxford OX1 3PU, UK. [13] Collaborative Innovation Center of Advanced Microstructures, 210093 Nanjing, China. [14] Shanghai Research Center for Quantum Sciences, 201315 Shanghai, China. [15]These authors contributed equally: Erjian Cheng, Wei Xia, Xianbiao Shi, Hongwei Fang. ✉email: liuzhk@shanghaitech.edu.cn; guoyf@shanghaitech.edu.cn; shiyan_li@fudan.edu.cn

Topological semimetals (TSMs), including Dirac, Weyl, nodal-line, and triple-point semimetals, can be divided into two categories—non-magnetic and magnetic TSMs depending on whether magnetism is involved[1–3]. Compared with better-known non-magnetic TSMs, magnetic TSMs have unique properties due to their broken time-reversal symmetry (TRS): for example, nonzero net Berry curvatures that can induce anomalous Hall or Nernst effects, only one pair of Weyl nodes for some magnetic Weyl semimetals, and a good ability to manipulate the spin for spintronics applications[3]. Moreover, when magnetism is involved, interactions of the external magnetic field with the magnetic moments can result in exotic properties, such as Weyl states induced by magnetic exchange[4,5]. However, in contrast to non-magnetic TSMs, theoretical predictions and experimental studies on magnetic TSMs are rarer and more difficult due to the complexity of the magnetic configuration for calculations and the difficulty in synthesis of single crystals[3]. In fact, the very nature of the interaction between magnetism and topology in magnetic TSMs remains mysterious. Given how few such compounds are known, seeking and fully characterizing new magnetic TSMs is a priority for the new light they may shed on these issues.

Recently, the non-magnetic $CaP_3$ family of materials was proposed as potential host of topological nodal-line (TNL) semimetals[6], among which $SrAs_3$ possesses a TNL feature at ambient pressure[7–9] and exotic properties under high pressure[10]. Isostructural with $SrAs_3$, $EuAs_3$ orders in an incommensurate antiferromagnetic (AFM) state at $T_N = 11$ K, and then undergoes an incommensurate–commensurate lock-in phase transition at $T_L = 10.3$ K, producing a collinear AFM ground state[11–16]. Previous electrical transport studies found an extremely anisotropic magnetoresistance (MR), which is strongly related to the magnetic configuration of $EuAs_3$[17]. However, experiments sensitive to the topology have not been reported on $EuAs_3$.

In this paper, we demonstrate a magnetism-induced topological transition from a TNL semimetal in the paramagnetic or the spin-polarized state to a topological massive Dirac metal in the AFM ground state. First, we explore the band structure in the AFM ground state through band calculations and transport measurements, demonstrating that $EuAs_3$ is a magnetic topological massive Dirac metal. Second, Shubnikov–de Haas (SdH) oscillations and band calculations in the spin-polarized state are displayed, yielding a proposal that $EuAs_3$ is a TNL semimetal with an extremely large magnetoresistance (XMR) of $\sim 2 \times 10^5$% at 1.8 K and 28.3 T. Third, our angle-resolved photoemission spectroscopy (ARPES) results in the paramagnetic state verify the nodal-line structure as predicted by band calculations. Ultimately, the origin of the XMR and a temperature-induced Lifshitz transition are revealed.

## Results

**Topological massive Dirac metal state in the AFM state.** $EuAs_3$ crystallizes in a monoclinic structure (space group $C2/m$, No. 12), and the magnetic moments of $Eu^{2+}$ are oriented parallel and antiparallel to the monoclinic $b$ axis[11–16], as plotted in Fig. 1a. Figure 1b shows the bulk and (110) surface Brillouin zones (BZs) of $EuAs_3$ in the doubled unit cell corresponding to its AFM ground state. The calculated band structure including spin–orbit coupling (SOC) in this magnetic ground state as determined by neutron diffraction experiments[12] for $EuAs_3$ is displayed in Fig. 1c. In addition to topological bands around the $\Gamma$ point, several trivial bands cross the Fermi level, indicating that $EuAs_3$ is a metal rather than a semimetal. In magnetic systems, TRS is broken. To preserve the Dirac node, extra symmetries, for example, the combination of inversion ($I$) and time-reversal ($T$) symmetries, i.e., $IT$, are necessary[3]. The Dirac band crossing is not

topologically protected, and it can be gapped out by SOC to turn into a gapped dispersion of massive Dirac fermions[3]. Following this clue, two massive Dirac points around $\Gamma$ point are identified, as shown in the inset to Fig. 1c. The complicated Fermi surface is composed of two hole sheets and one electron sheet in the AFM state (Fig. 1d), all of three-dimensional (3D) character. The electron sheet consists of two individual pockets, i.e., electron_1 and electron_2, which can both be detected by quantum oscillations.

Projected band structure analysis shows that the low energy states near the Fermi level are dominated by As-$4p$ states (Fig. 1e). There are clear signatures of band inversion between As-$p_{x,y}$ and As-$p_z$ orbitals at the $\Gamma$ point. To identify the topological character, we calculated the $Z_2$ invariant by employing the Willson loop method[18]. Figure 1f, g show the evolution of the Wannier charge center on two representative planes of the bulk BZ. From the calculations, the $Z_2$ invariant for the $k_z = 0$ plane is 1, whereas $Z_2$ is 0 for the $k_z = 0.5$ plane, providing strong evidence for nontrivial topology. Moreover, topologically protected surface states are expected, and we can unambiguously identify nontrivial surface states in the surface spectrum for the semi-infinite (010) surface, as displayed in Fig. 1h, confirming further the nontrivial topology in the AFM state.

To verify the predictions from band calculations, we conducted electrical transport measurements. Resistivity in zero magnetic field is plotted in Fig. 2a, which displays typical metallic behavior with a low-temperature peak corresponding to the magnetic transitions. The magnetic transitions were also found by thermodynamic measurements (Supplementary Fig. 1) to be consistent with previous reports[11–16]. The inset to Fig. 2a shows the fit to the resistivity data below 2.5 K using a power law: $\rho = \rho_0 + AT^2$, where $\rho_0$ is the residual resistivity and $A$ the electronic scattering coefficient. The fit gives a residual resistivity $\rho_0$ of 2.6 μΩcm, and the residual resistivity ratio $\rho_{300K}/\rho_0$ is ~72. Figure 2b shows the low-field MR data with evident SdH oscillations. $B_M$ in Fig. 2b denotes the critical field, above which the spins are fully polarized by the external magnetic field. The SdH oscillation amplitude can be described by the Lifshitz–Kosevich formula[1,2]: $\Delta \rho_{xx} \frac{2\pi^2 k_B T/\hbar \omega_c}{\sinh(2\pi^2 k_B T/\hbar \omega_c)} e^{2\pi^2 k_B T_D/\hbar \omega_c} \cos 2\pi (\frac{F}{B} - \gamma + \delta)$, where $\omega_c = eB/m^*$ is the cyclotron frequency and $T_D$ is the Dingle temperature. $\gamma = \frac{1}{2} - (\frac{1}{2\pi})\phi_B$ ($0 \leq \gamma \leq 1$) is the Onsager phase factor, and $\phi_B$ is a geometrical phase known as the Berry phase. For a topological system with peculiar electron state degeneracy and intra-band coupling, a $\pi$ Berry phase will be observed. $2\pi\delta$ is an additional phase shift resulting from the curvature of the Fermi surface in the third direction, where $\delta$ varies from 0 to $\pm 1/8$ for a quasi-two-dimensional (quasi-2D) cylindrical Fermi surface and a corrugated 3D Fermi surface, respectively[7,8]. The cyclotron effective mass $m^*$ can be obtained from the thermal damping factor $R_T = \frac{2\pi^2 k_B T/\hbar \omega_c}{\sinh(2\pi^2 k_B T/\hbar \omega_c)}$.

By analyzing the oscillatory component (inset to Fig. 2c) below $B_M$ via fast Fourier transform (FFT), four bands are uncovered, i.e., 156, 185, 217, and 7 T, referred to as $\alpha$, $\beta$, $\gamma^1$, and $\gamma^2$, respectively, in line with the band calculations. To check their topological nature, a Landau index fan diagram is plotted in Fig. 2d, yielding intercepts of 0.6(2), 0.5(1), −0.03(9), and 0.07(8) for $\alpha$, $\beta$, $\gamma^1$, and $\gamma^2$, respectively. Throughout this paper, we assign integer indices to the $\Delta \rho_{xx}$ peak positions in $1/B$ and half integer indices to the $\Delta \rho_{xx}$ valleys. According to the Lifshitz–Onsager quantization rule for a corrugated 3D Fermi surface, intercepts falling between −1/8 and 1/8 suggest nonzero Berry phase, while intercepts in the range 3/8–5/8 indicate trivial band topology. Therefore, the $\gamma^1$ and $\gamma^2$ bands may be topologically protected, while the other two are trivial. However, the index number is >20 ($\alpha$, $\beta$, and $\gamma^1$ pockets), and hence the extrapolation from the Landau fan plots may have biggish uncertainty. In order to

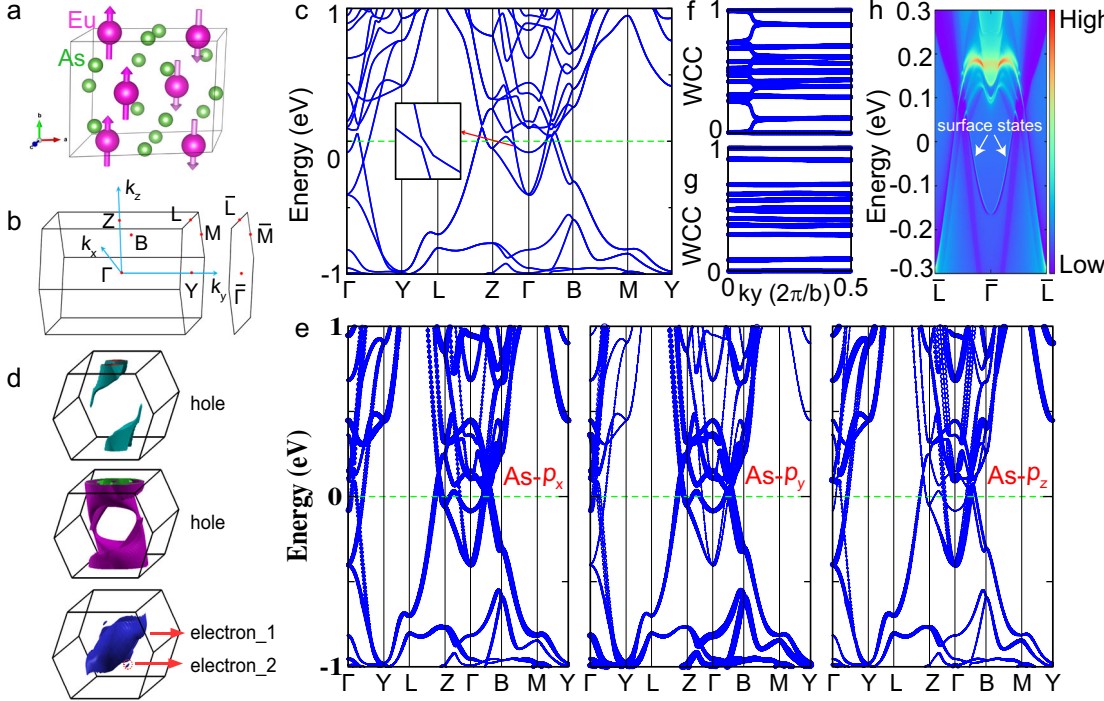

**Fig. 1 Topological massive Dirac metal state in the antiferromagnetic state of EuAs$_3$, revealed by band calculations. a** Schematic for the crystal structure of EuAs$_3$ in the doubled magnetic unit cell. The arrows on Eu$^{2+}$ represent the spin directions, which are parallel and antiparallel to the *b* axis. **b** Bulk and (110)-projected surface BZs of the doubled magnetic unit cell of EuAs$_3$ with several high-symmetry points marked. **c** Band structure of EuAs$_3$ from GGA + SOC + U ($U = 5$ eV) calculations for the AFM ground state. The inset shows the massive Dirac point with a small gap. **d** Fermi surfaces of EuAs$_3$ derived from the band structure. **e** Projected band structure of EuAs$_3$, where the symbol size represents the projected weight of Bloch states onto the As-$p_x$, $p_y$, and $p_z$ orbitals as labeled. Band inversion can be observed at the Γ point. The Wannier charge center is calculated in the **f** $k_z = 0$ and **g** $k_z = 0.5$ planes. **h** Calculated surface states on the (010) surface. The nontrivial topological surface states are clearly visible.

validate the topological nature, more solid evidences are needed. Other parameters for these four bands, such as the Fermi energy $E_F$, extremal cross-sectional areas $A_F$, Fermi momentum $k_F$, Fermi velocity $v_F$, cyclotron effective mass $m^*$, and Dingle temperature $T_D$, are calculated and summarized in Table 1.

In TSMs, in addition to nonzero Berry phase, the n-LMR induced by the chiral anomaly can also serve as a smoking gun for nontrivial topology[19–21]. Figure 2e displays the n-LMR of EuAs$_3$ with magnetic field parallel to the electric current I. The kinks in the n-LMR curves below the ordering temperature arise from the field-induced transitions[15–17], i.e., from a collinear AFM phase to incommensurate and commensurate spiral phases[15–17]. Negative MR in magnetic systems is not uncommon[19–21] when the applied magnetic field suppresses the inelastic magnetic scattering from local moments or magnetic impurities, leading to a negative MR for charge transport along all directions[19–21]. However, in EuAs$_3$ we only observed a n-LMR when the magnetic field is applied parallel to the electric current (Supplementary Fig. 2). Furthermore, if the applied external magnetic field has a strong effect on the magnetic scattering and induces a n-LMR, the changes in the n-LMR will occur predominantly below and above the ordering temperature. This is not observed. Instead, we find several minor kinks arising from the magnetic transitions, on top of a much larger signal. Therefore, the suppression of magnetic scattering can be excluded as the origin of the n-LMR in EuAs$_3$. The n-LMR also displays a wide variety of temperature dependences, ruling out current jetting effects and the weak localization effect[19–21]. Since nontrivial band topology has been suggested in EuAs$_3$, the chiral anomaly arising from Weyl fermions is the most likely mechanism behind the n-LMR.

The n-LMR induced by the chiral anomaly in TSMs can be analyzed through the Adler–Bell–Jackiw (ABJ) chiral anomaly equation[19–21]: $\sigma(B) = (1 + C_w B^2)(\sigma_0 + a\sqrt{B}) + \sigma_N$, where $\sigma_0$, $C_w$, and $\sigma_N^{-1} = \rho_0 + AB^2$ denote the conductivity at zero field, a temperature-dependent positive parameter originating from chiral anomaly, and the conventional nonlinear band contribution around Fermi energy, respectively. Figure 2f shows the conductivity and the fit to the data below 3 T at various temperatures. The data above the ordering temperature are well described by the ABJ equation, while the data at lower temperatures do not fit as well, which may be ascribed to magnetic transitions or topological transitions. The inset to Fig. 2f shows the temperature dependence of $C_w$. At 2 K, $C_w$ is 0.253(7) T$^{-2}$. With increasing temperature, a clear anomaly in $C_w$ around the ordering temperature can be observed, verifying the proposal above. When $T > T_N$, $C_w$ decreases monotonically, as observed in SrAs$_3$[7]. Taken together, these results demonstrate that EuAs$_3$ is a magnetic topological massive Dirac metal in its AFM ground state.

**Topological state in the spin-polarized state.** We now turn to the exploration of topology in the spin-polarized state, where in Fig. 2b we have already observed a clear change in the quantum oscillations. Figure 3a plots the MR of EuAs$_3$ in higher magnetic field, and an unsaturated XMR ~2 × 10$^5$% at 1.8 K and 28.3 T is observed. By analyzing the oscillatory components above $B_M$ (inset in Fig. 3b), frequency components are identified at $F = 93, 158, 346$, and 597 T, which are referred to here as the ξ, α′, ε, and η bands, respectively (Fig. 3b). These four bands are different from those in lower field (Fig. 2c), indicating that they are likely rooted in different band structure. This is unsurprising since the unit cell is no longer doubled

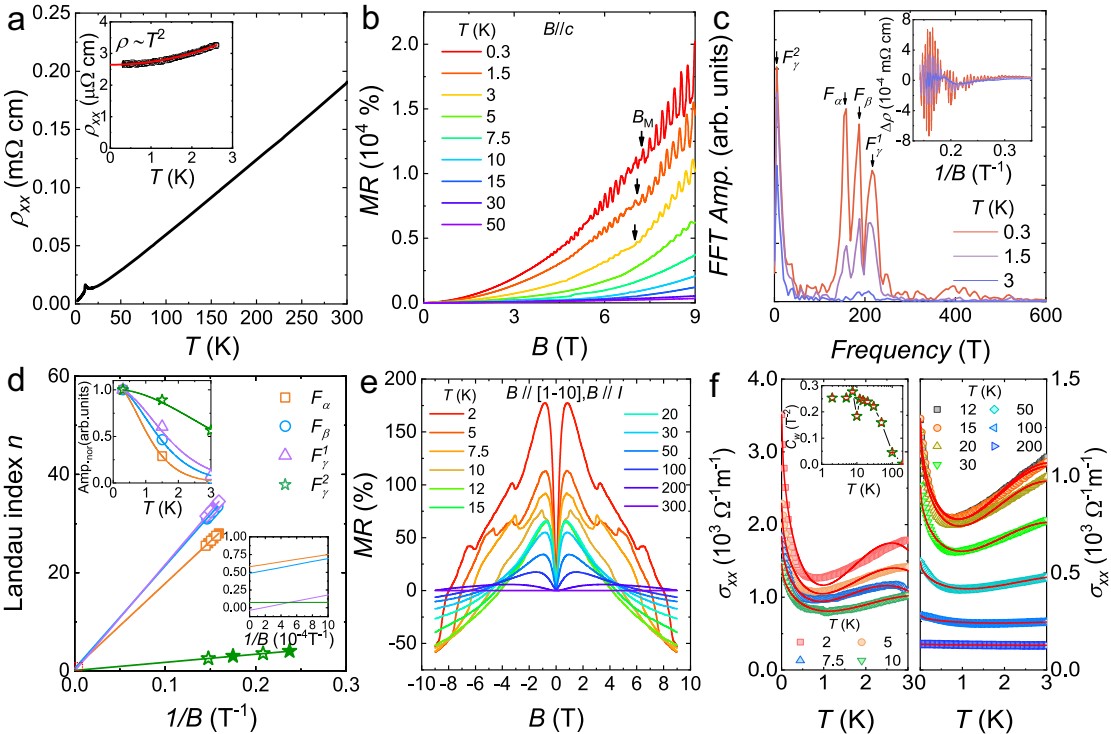

**Fig. 2 Quantum oscillations and negative longitudinal magnetoresistance (*n*-LMR) in the antiferromagnetic state of EuAs₃.** **a** Resistivity of EuAs₃ single crystal in zero magnetic field. The inset shows the fit to the low-temperature data. **b** MR accompanied by distinct SdH oscillations. $B_M$ represents the critical magnetic field, which induces a magnetic transition from a collinear antiferromagnetic phase to a polarized ferromagnetic phase. **c** FFT results at various temperatures. The inset displays the oscillatory component $\rho_{xx}$ below $B_M$. Four bands, i.e., $\alpha$, $\beta$, $\gamma^1$, and $\gamma^2$, can be distinguished. The latter two construct one electron sheet. **d** Landau index *n* plotted against $1/B$ for the SdH oscillations at 0.3 K. Lines represent linear fits. The right inset shows the extrapolation of $1/B$ to zero. The left inset shows the normalized FFT amplitude (Amp.$_{nor}$) as a function of temperature, and the solid lines represent fits to the Lifshitz–Kosevich formula. **e** *n*-LMR measured with magnetic field parallel to the electric current *I* at various temperatures. **f** Longitudinal conductivity at various temperatures fit to the Adler–Bell–Jackiw chiral anomaly equation. The inset shows the emergence of a positive parameter originating from the chiral anomaly $C_w$, and error bar of the data is determined from fitting.

**Table 1 Parameters derived from quantum oscillations in different magnetic field range for EuAs₃.**

| Magnetic field range (T) | | F (T) | $E_F$ (meV) | $A_F$ ($10^{-3}$ Å$^{-2}$) | $k_F$ ($10^{-2}$ Å$^{-1}$) | $v_F$ ($10^5$ m/s) | $m^*$ ($m_0$) | $T_D$ (K) |
|---|---|---|---|---|---|---|---|---|
| 2.85–7 (AFM state) | $\alpha$ | 156 | 0.6 | 14.9 | 6.9 | 1.3 | 0.580 (1) | 15.2 (4) |
| | $\beta$ | 185 | 0.8 | 17.7 | 7.5 | 1.8 | 0.45 (1) | 5.0 (6) |
| | $\gamma^1$ | 217 | 1.0 | 20.8 | 8.1 | 2.3 | 0.38 (3) | 8 (1) |
| | $\gamma^2$ | 7 | 2.1 | 0.3 | 1.0 | 0.6 | 0.178 (6) | 8.0 (5) |
| 11.1–28.3 (spin-polarized state) | $\xi$ | 93 | 4.0 | 8.9 | 5.3 | 1.7 | 0.37 (1) | 12 (2) |
| | $\alpha'$ | 158 | 3.1 | 15.2 | 6.9 | 1.6 | 0.51 (1) | 5.9 (1) |
| | $\varepsilon$ | 346 | 4.5 | 33.3 | 10.3 | 3.6 | 0.329 (4) | 13.1 (6) |
| | $\eta$ | 597 | 4.0 | 57.4 | 13.5 | 4.2 | 0.370 (3) | 9.8 (7) |

*F*, $E_F$, $A_F$, $k_F$, $v_F$, $m^*$, and $T_D$ represent FFT frequency, Fermi energy, extremal cross-sectional areas of Fermi surface, Fermi momentum, Fermi velocity, cyclotron effective mass, and Dingle temperature, respectively.

by antiferromagnetism, but the field-induced spin polarization can also play a significant role. We thus conducted band calculations for the field-polarized state (Supplementary Fig. 3) and the paramagnetic state (Supplementary Fig. 4), and these are indeed quite different, as we discuss in more detail in Supplementary Notes 3 and 4.

In the field-polarized state, we find four Fermi surface sheets—two electron and two hole sheets and double nodal loops at the *Y* point, one each for the spin-up and spin-down channels (Supplementary Fig. 3d). To identify the topological nature of the four bands seen in quantum oscillations, a Landau index fan diagram is plotted in Fig. 3c, and the intercepts are −0.0(1), 0.67(3), 0.34(4), and 0.61(4) for the $\xi$, $\alpha'$, $\varepsilon$, and $\eta$ bands,

respectively. Although the intercepts from the fit with large Landau index number cannot serve as a smoking gun for topology, we still use them, because it is difficult for us to evaluate the topological nature in the fully spin-polarized state above 11.0 T (the critical field is deduced from specific heat in Supplementary Fig. 1c). The intercepts give a hint that the $\xi$ band is topologically protected, while the $\alpha'$ and $\eta$ bands are topologically trivial. The intermediate value for $\varepsilon$ is suggestive of a possible nontrivial Berry phase but does not allow a strong conclusion and will require further verification. The cyclotron effective masses $m^*$ for these four pockets can be obtained by fitting the temperature dependence of the normalized FFT

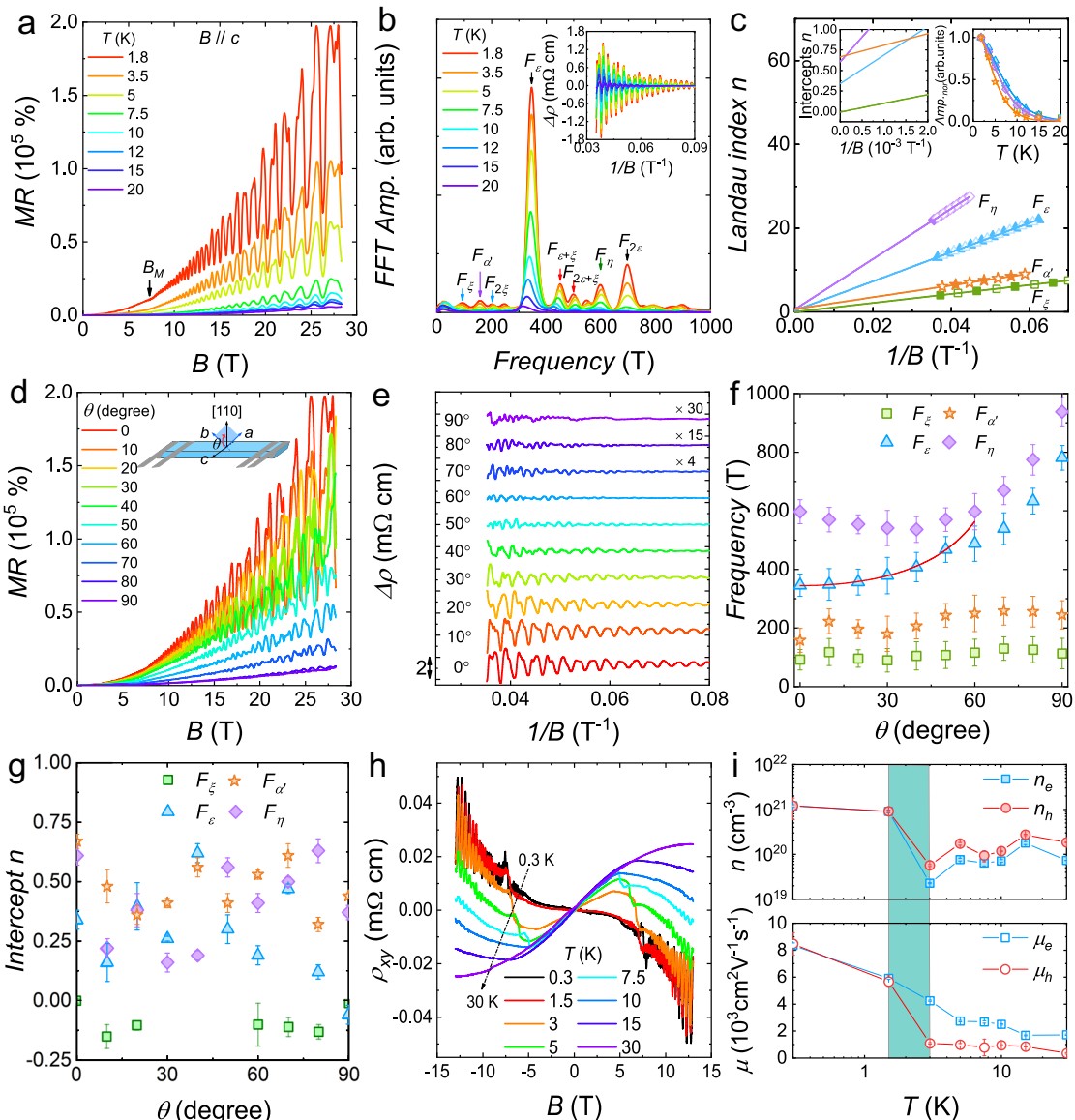

**Fig. 3 Quantum oscillation study in the spin-polarized state of EuAs₃ and Hall resistivity measurements. a** Magnetoresistance measurements of EuAs₃ single crystal under higher magnetic field up to 28.3 T. **b** FFT results at various temperatures, yielding the four bands $\xi$, $\alpha'$, $\varepsilon$, and $\eta$. The inset displays the oscillatory component $\rho_{xx}$ above $B_M$. **c** Landau index $n$ plotted against $1/B$ for the SdH oscillations at 1.8 K. The left inset shows the extrapolation of $1/B$ to zero. The right inset shows the normalized FFT amplitude (Amp.ₙₒᵣ) as a function of temperature, and the solid lines represent the Lifshitz–Kosevich formula fit. **d** SdH oscillations at different angles; the inset is a schematic illustration of the experimental geometry and the angle $\theta$. For $\theta = 0°$, the magnetic field is parallel to the $c$ axis. For $\theta = 90°$, the magnetic field is applied along the [110] direction. **e** The oscillatory component $\rho_{xx}$ as a function of $1/B$. Angular dependence of **f** the FFT frequencies, where error bars represent the full widths at half maximum of the FFT peaks, and **g** the Landau level index intercepts. The error bar of the intercepts is determined from fitting. **h** Hall resistivity at various temperatures. **i** Carrier concentration and mobility as a function of temperature, and error bar is determined from fitting. The shadow area represents the temperature interval where a Lifshitz transition takes place.

amplitude, as shown in the right inset to Fig. 3c. Other parameters can be also extracted, and all values are summarized in Table 1.

To better reveal the Fermi surface anisotropy and topology of EuAs₃, angle-dependent MR measurements have been performed at 1.8 K, in the experimental geometry shown in the inset to Fig. 3d. Upon rotating sample from 0° to 90°, the magnitude of the MR is reduced monotonically, as also seen in a polar plot of the MR (Supplementary Fig. 2b). We extract the frequency components for the $\xi$, $\alpha'$, $\varepsilon$, and $\eta$ bands by analyzing the oscillatory component (Fig. 3e) and summarize the results in

Fig. 3f. The angle dependence of the $\xi$, $\alpha'$, and $\eta$ bands is of 3D character, while the $\varepsilon$ band is well described below 50° by the formula $F = F_{3D} + F_{2D}/\cos(\theta)$, where $F_{2D}$ and $F_{3D}$ denote 2D and 3D components, respectively. The ratio between the 2D and 3D components derived from the fit ($F_{2D}/F_{3D}$) is ~1.76, suggesting that the $\varepsilon$ pocket exhibits mainly 2D character although a 3D component also exists.

Now, we turn to the angle dependence of the Berry phase. As shown in Fig. 3g, the intercept for the topological $\xi$ band shows strong angle dependence, similar to results in other systems such as Cd₃As₂[22], ZrSiM (M = Se, Te)[23], or ZrTe₅[24]. For $\theta < 30°$ and

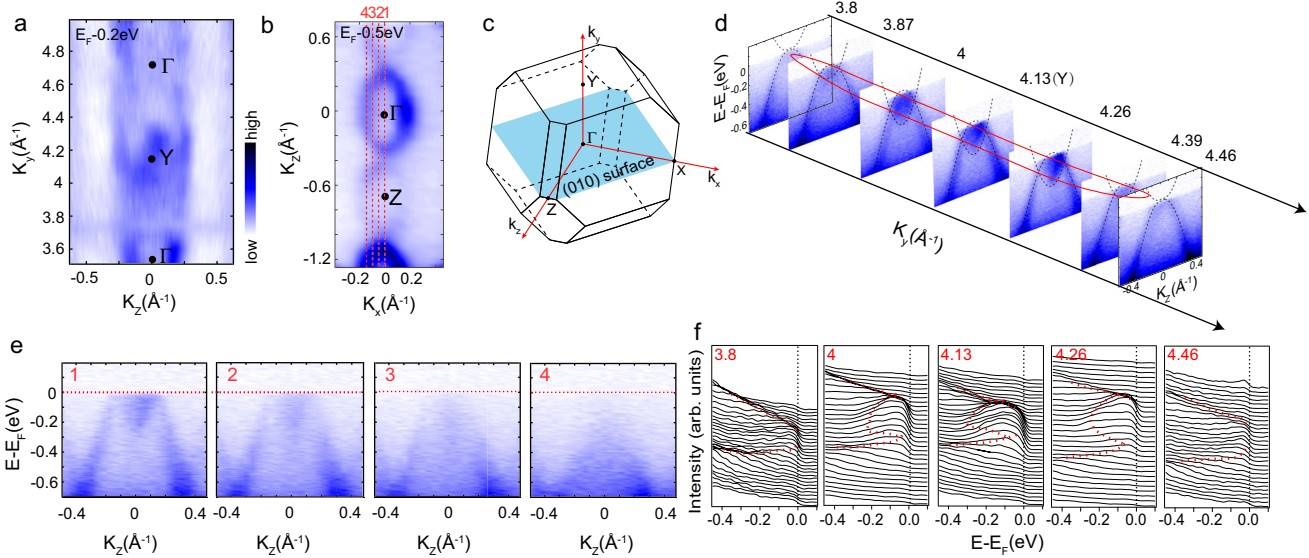

**Fig. 4 Verification of topological nodal-line structure by ARPES measurements in the paramagnetic state of EuAs₃. a** Photon energy-dependent plot of photoemission intensity in the $k_z$–$k_y$ plane taken at $E_F$ −0.2 eV. **b** Photoemission intensity map of constant energy contours at 0.5 eV below $E_F$ in the $k_x$–$k_z$ plane, the data were collected using photons with $h\nu = 55$ eV. **c** The Brillouin zone (BZ) of EuAs₃, with high-symmetry points and (010) surface labeled. **d** The band dispersions along $k_y$ direction probed by different photon energies. The calculations plotted by the black dotted curves superimposed on the experimental electronic structure. The red ellipse illustrates the topological nontrivial nodal loop schematically. **e** The band dispersions along cuts 1–4 as indicated in **b**, respectively. **f** Corresponding energy-distribution curves (EDCs) taken at different photon energies.

$\theta \geq 60°$, the intercept falls between −1/8 and 1/8, while it falls between 3/8 and 5/8 for $30° \leq \theta < 60°$. For the $\varepsilon$ band, the intercept from 0° to 70° fluctuates between 1/8 and 5/8, averaging to 0.34(5), which is suggestive of trivial topology. However, when $\theta$ reaches 80° and 90°, this intercept falls between −1/8 and 1/8, implying nonzero Berry phase. For $\eta$ and $\alpha'$ bands, the intercepts at all angles remain between 1/8 and 5/8, averaging 0.4(2) and 0.5(2), respectively, indicating trivial topology. Figure 3h, i show Hall results, which will be discussed later.

**TNL structure in the paramagnetic state.** Since our band structure calculations identify nodal loops at $Y$ and our quantum oscillation data indicate nontrivial band topology, we also directly investigated the band structure with ARPES (Fig. 4). Momentum analysis in this technique is incompatible with magnetic field, so we investigated the paramagnetic rather than the field-polarized state; however, as shown in more detail in Supplementary Figs. 3 and 4, a closed nodal loop persists at the $Y$ point in the paramagnetic state. In order to visualize the nodal loop in EuAs₃, the photon energy dependence of the electronic structure along the $k_y$ direction was investigated at 12 K within the vertical plane of the (010) cleaved surface, as sketched in Fig. 4c.

From the intensity plot of the Fermi surface at 12 K in the $k_y$–$k_z$ plane (Fig. 4a) taken at $E_F$ −0.2 eV, the pocket centered at the $Y$ point (54 eV) can be easily identified, and two nodes arising from the crossing of the electron-like and hole-like bands can be also observed in Fig. 4b, which agrees with the band calculations. As observed in SrAs₃, the drumhead-like surface state of EuAs₃ is buried in the bulk state, so it cannot be resolved by ARPES. For ARPES cuts away from the $Y$ point, the band-crossing area shrinks gradually and finally disappears, and the topological nontrivial nodal loop encircled the $Y$ point, illustrated by the red ellipse in Fig. 4d. Besides, the $k_y$-dependent evolution of the band structure shows a good agreement with the calculations (the black dotted curves in Fig. 4d) and the corresponding energy-distribution curves could further confirm the nodes introduced by the band crossing and their $k_y$-dependent evolution in Fig. 4f

(dashed line is a guide for the eyes to trace the dispersions). The evolution of the nodes along the $k_x$-direction is presented in Fig. 4e, which shows the band dispersions along cuts 1–4 indicated in Fig. 4b (photoemission intensity map of constant energy contours at 0.5 eV below $E_F$). We noticed both the electron- and hole-like bands deplete their spectral weight from 1 to 4, consistent with the band-crossing scenario. However, whether or not a gap opens in the nodes away from $k_x = 0$ remains vague due to the intrinsic broadness of the electron-like band. We also measure the electronic structure of another sample at 18 K, as shown in Supplementary Fig. 5, and obtain the same results. And we also estimate the Fermi momenta $k_F$ and Fermi velocities $v_F$ to be $k_F = 0.12$ and 0.14 Å⁻¹ and $v_F = 3.7 \times 10^5$ and $1.17 \times 10^5$ m/s, respectively, for the hole and electron bands, the same order of magnitude as for the $\varepsilon$ and $\eta$ bands (see Table 1). These data are extremely similar to what was found in SrAs₃[9].

The verification of the nodal-line structure in the paramagnetic state by utilizing ARPES measurements and density functional theory (DFT) calculations shows remarkable agreement between the theoretical and experimental values. For the spin-polarized state, which is predicted to hosts closely similar but spin–split band structure to the paramagnetic state, nodal-line structure is thus strongly expected to exist in spin-polarized EuAs₃. Very recently, lifted degeneracy of the Bloch bands was observed in the paramagnetic phase of EuCd₂As₂, producing a spin-fluctuation-induced Weyl semimetal state[25]. The magnetic susceptibility in EuAs₃ reveals a positive Curie–Weiss temperature $T_{CW}$ of 4.4 K for magnetic fields applied within the $ab$ plane (Supplementary Fig. 6), suggestive of ferromagnetic fluctuations deep in the paramagnetic phase. The ferromagnetic correlations in EuAs₃ may induce band splitting within the paramagnetic phase, which may be resolvable with higher-resolution ARPES, such as laser ARPES.

## Discussion

It is clear from our transport measurements that the electronic structure in the antiferromagnetically ordered state is very different from that found in the field-polarized or paramagnetic states. This is

a consequence of the doubling of the unit cell due to AFM order and the coupling of this magnetic order to the electronic structure and is well described by our band calculations. However, a possible additional Lifshitz transition below 3 K has also been suggested.

Figure 3h shows the Hall resistivity ($\rho_{xy}$) from 0.3 to 30 K. The $\rho_{xy}$ curves are clearly nonlinear, implying the coexistence of two types of carriers, as predicted by band structure calculations. On cooling, the slope of the curve changes from positive to negative, indicating an increased contribution from electron carriers. The carrier concentration and mobility are extracted by fitting the low-field Hall conductivity with a two-carrier model, and results are summarized in Fig. 3i. For $3 \leq T \leq 30$ K, the concentration of hole carriers is larger than that of electron carriers. Upon decreasing the temperature <3 K, the concentration of electron carriers is suddenly enhanced, accompanied by a sharp increase in the mobility of hole carriers. These indicate a possible Lifshitz transition.

Temperature-induced Lifshitz transitions are also observed in other TSMs, for example, MTe$_5$ (M = Zr, Hf)[26,27], InTe$_{1-\delta}$[28], ZrSiSe[29], WTe$_2$[30], and TaIrTe$_4$[31]. Anomalies in both longitudinal resistivity and Hall resistivity/coefficient can be found in MTe$_5$ (M = Zr, Hf)[26,27], InTe$_{1-\delta}$[28], and ZrSiSe[29], but not in WTe$_2$[30] and TaIrTe$_4$[31]. Since we have observed the change of Hall resistivity in EuAs$_3$ (Fig. 3h), we wonder how the longitudinal resistivity evolves with decreasing temperature. The inset of Fig. 2a shows the low-temperature resistivity from 0.3 to 2.5 K in zero field, and we did not observe any distinct anomaly. Considering that the variations in resistivity may be very weak and the temperature range from 0.3 to 2.5 K is not appropriate, we measured two more samples (denoted as Sample 4 and Sample 5 in Supplementary Fig. 7a), and found that there is a very weak anomaly at ~2.3 K in resistivity for both samples (see Supplementary Fig. 7b). Since the Lifshitz transition should also manifest in quantum oscillations, we further check the low-field MR data below $B_M$ in Fig. 2b and the FFT in Fig. 2c. One can see that, due to the limited oscillatory periods and/or noise at 3 K, the Lifshitz transition cannot be resolved from the low-field quantum oscillation data. To verify it, other low-temperature probes are needed, for example ARPES and scanning tunneling microscopy/scanning tunneling spectroscopy.

Now, we turn to the high-field state >$B_M$. The temperature dependence of Hall coefficient measured at 9 T for Sample 5 is shown in Supplementary Fig. 7c. With decreasing temperature, a small peak at ~3.6 K arises and Hall coefficient changes its sign from positive to negative at ~2.3 K. We then analyze the oscillatory component ($\Delta\rho_{xy}$) > $B_M$, and a new oscillation frequency of 374 T (denoted as the $\varphi$ band) can be clearly distinguished at 0.3 K, as shown in Supplementary Fig. 8a. The trivial topology nature for $\varphi$ band has also been demonstrated (see Supplementary Fig. 8b). Therefore, a temperature-induced Lifshitz transition likely exists in both AFM and spin-polarized states, although the change of Fermi surface topology with temperature in these two states may be different. Generally speaking, Lifshitz transitions are related to electronic transitions at zero temperature and involve abrupt changes of the Fermi surface topology. However, in topological materials, Lifshitz transitions can also involve other types of zero-energy modes, such as Weyl or Dirac nodes, nodal lines, flat bands, Majorana modes, etc.[32]. It has been proposed that multiple types of novel Lifshitz transitions involving Weyl points are possible depending on how they connect Fermi surfaces and pockets. For instance, the Lifshitz transition can correspond to the transfer of Berry flux between Fermi pockets connected by type-II Weyl points[33]. To understand the physics behind the low-temperature Lifshitz transition in EuAs$_3$, more work is needed.

According to the conventional charge-carrier compensation picture for XMR, the ratio $n_h/n_e$ should be unity[1,2]. At 0.3 K, $n_h/n_e$ for EuAs$_3$ is ~1.0, consistent with this picture. However, for $3 \leq T \leq 30$ K, $n_h/n_e$ varies between 1.5 and 2.5 while the MR

remains large and unsaturated, evidently excluding the charge-compensation picture for EuAs$_3$. XMR is also frequently encountered in the cases of topologically protected electronic band structure and when open orbits contribute[1,2,34,35]. According to the open-orbit effect, the unsaturated XMR is only observed for current along the open orbits[35]. However, the observation of the unsaturated XMR with different current direction in EuAs$_3$ excludes the open-orbit effect (see Supplementary Fig. 9). Besides, for both the charge-carrier compensation picture and open-orbit effect, a $B^2$ dependence of MR is suggested[34,35], which is different from the situation of EuAs$_3$ reported here (Supplementary Fig. 9c). Since we have verified nontrivial band topology in EuAs$_3$, we consider this the more likely explanation.

In summary, combining band calculations, electrical transport, and ARPES measurements on the magnetic compound EuAs$_3$, we report a magnetism-induced topological transition from a TNL semimetal in the paramagnetic or the spin-polarized state to a topological massive Dirac metal in the AFM ground state. The paramagnetic and spin-polarized states differ by the splitting of a topological nodal line associated with the spin splitting of the band structure. An XMR of ~$2 \times 10^5$% and an as-yet-unexplained temperature-induced Lifshitz transition <3 K have also been revealed. These results indicate that magnetic EuAs$_3$ could serve as a unique platform to explore exotic physics at the interface of magnetism and topology.

## Methods

**Sample synthesis.** Eu (99.95%, Alfa Aesar), As (99.999%, PrMat), and Bi (99.9999%, Aladdin) blocks were mixed in a molar ratio of 1:3:26 and placed into an alumina crucible. The crucible was sealed in a quartz ampoule under vacuum and subsequently heated to 900 °C in 15 h. After reaction at this temperature for 20 h, the ampoule was cooled to 700 °C over 20 h and then slowly cooled to 450 °C at −1 °C/h. The excess Bi flux was then removed using a centrifuge, revealing EuAs$_3$ single crystals with black shiny metallic luster.

**Electrical transport and thermodynamic measurements.** For electrical transport measurements, a single crystal was cut into a bar shape. A standard four-probe method was used for the longitudinal resistivity measurement. Data were collected in a $^3$He and a $^4$He cryostat. Magnetic susceptibility and specific heat measurements were performed in a magnetic property measurement system (MPMS, Quantum Design) and a physical property measurement system (PPMS, Quantum Design), respectively. High-field measurements were performed at the Steady High Magnetic Field Facilities, High Magnetic Field Laboratory, Chinese Academy of Sciences in Hefei.

**ARPES measurements.** ARPES measurements were performed at beam line BL13U at the National Synchrotron Radiation Laboratory (NSRL), China (photon energy $hv = 12$–38 eV); beam line BL03U of Shanghai Synchrotron Radiation Facility (SSRF), China (photon energy $hv = 34$–90 eV). The samples were cleaved in situ at 18 K (12 K) and measured in ultrahigh vacuum with a base pressure of better than $3.5 \times 10^{-11}$ ($5 \times 10^{-11}$) mbar at NSRL (SSRF). Data were recorded by a Scienta R4000 at NSRL and SSRF. The energy and momentum resolution were 10 meV and 0.2°, respectively.

**DFT calculations.** First-principles calculations were carried out within the framework of the projector augmented wave method[36,37] and employed the generalized gradient approximation (GGA)[38] with Perdew–Burke–Ernzerhof formula[39], as implemented in the Vienna ab initio Simulation Package[40]. Two unit cells repeated along the *b* axis were adopted to simulate the AFM configuration indicated by neutron diffraction experiment for EuAs$_3$[14]. The energy cutoff was chosen to be 500 eV. A Γ-centered $8 \times 6 \times 14$ Monkhorst–Pack *k*-point grid was used to produce the well-converged results for the AFM phase. For the spin-polarized and paramagnetic band calculations, the same unit cell was used. Γ-centered $10 \times 10 \times 10$ and $6 \times 6 \times 6$ grids were used in the first BZ for the unit cell and supercell magnetic structures, respectively. The convergence criterion of energy in relaxation was set to be $10^{-6}$ eV and the atomic positions were fully relaxed until the maximum force on each atom was <0.02 eV/Å. The electronic correlations of Eu-4*f* states were treated by the GGA + *U* method[41]. SOC was considered in a self-consistent manner. The Wannier90 package[42] was adopted to construct Wannier functions from the first-principles results. The WannierTools code[43] was used to investigate the topological features of surface state spectra.

## Data availability

The data that support the findings of this study are available from the corresponding author upon reasonable request.

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

## Acknowledgements

This work is supported by the Natural Science Foundation of China (Grant Nos. 12034004, 11674367, 11674229, and 11874264), the Ministry of Science and Technology of China (Grant Nos. 2016YFA0300503 and 2017YFA0305400), the Shanghai Municipal Science and Technology Major Project (Grant No. 2019SHZDZX01), the Zhejiang Provincial Natural Science Foundation (Grant No. LZ18A040002), and the Ningbo Science and Technology Bureau (Grant No. 2018B10060). W.Z. is supported by the Shenzhen Peacock Team Plan (KQTD20170809110344233) and Bureau of Industry and Information Technology of Shenzhen through the Graphene Manufacturing Innovation Center (201901161514). Y.G. acknowledges research funds from the State Key Laboratory of Surface Physics and Department of Physics, Fudan University (Grant No. KF2019 06). C.X. was supported by the Users with Excellence Project of Hefei Science Center CAS (Grant No. 2018HSC-UE015). D.C.P. is supported by the Chinese Academy of Sciences through 2018PM0036 and from the Deutsche Forschungsgemeinschaft (DFG), through project C03 of the Collaborative Research Center SFB 1143 (project-ID 247310070). The authors are grateful for support from the Analytical Instrumentation Center (# SPST-AIC10112914), SPST, ShanghaiTech University. Part of this research used beamline 03U of the Shanghai Synchrotron Radiation Facility, which is supported by ME2 project under contract No. 11227902 from the National Natural Science Foundation of China.

## Author contributions

S.L. and Y.G. conceived the idea and designed the experiments. E.C. was responsible for electronic transport experiments. W.X., X.W., N.Y., and H.S. performed sample synthesis and partial data analysis. X.S., S.X., W.R., and W.Z. performed the electronic band calculations. C.X., L.W., and L.P. helped with the MR measurements in Hefei. H.F., C.W., Y.C., and Z.L. performed ARPES measurements and analysis. Z.L., Y.G., and S.L. supervised the project. E.C., D.C.P., Y.G., and S.L. analyzed the data and wrote the paper. All authors discussed the results and commented on the manuscript.

## Competing interests

The authors declare no competing interests.
