## [Peer Review File · Nature Communications]

Reviewers' comments:

Reviewer #1 (Remarks to the Author):

The manuscript reported the ab initio band calculations, electrical transport and angle-resolved photoemission spectroscopy (ARPES) measurements on the magnetic semimetal EuAs₃. The key point is that they demonstrated a magnetism-induced topological transition from a topological nodal-line semimetal in the paramagnetic or the spin-polarized state to a topological massive Dirac metal in the antiferromagnetic (AFM) ground state at low temperature. Also, a temperature-induced Lifshitz transition accompanied by the emergence of a new band below 3 K is revealed. All of these results are interesting. However, except for the evidence of the topological nodal-line semimetal in paramagnetic state can be recognized by ARPES, which is corroborated by band calculation, other assignments or statements actually still remain uncertainty.

1)The high field Shubnikov-de Haas (SdH) data at different temperatures sounds great, it presents very strong quantum oscillations in the spin polarized state. From Fig. 3a and 3b, the oscillation patterns at 15 K which is above the AFM transition T_n sound the same with those in the spin polarized state below T_n , I guess this is why the authors claim a magnetism-induced topological transition from a topological nodal-line semimetal in the spin-polarized state to a topological massive Dirac metal in the antiferromagnetic (AFM) ground state at low temperature. My concerning is that: is the band structure in this spin polarized state below T_n actually the same with that in the paramagnetic state above T_n (eg., the Nodal-line structures at the Y point in the Brillouin zone are proposed in both the spin-polarized and paramagnetic states)? In Figure 3h, the Hall data shows a hole-dominated state in paramagnetic state, but electron-dominated state in AFM or spin-polarized state below T_n .

2)In Figure 3 a, the quantum oscillations are well resolved over the full magnetic field range in the measured temperatures, why the authors only treat the SDH data in the high-field range above B_m , while the low field data was selected from PPMS measurements. Maybe the low field data is noisy, do the authors try to make FFT from Figure 3a in the full field range? Do they find 8 different frequencies or only 4 frequencies?

3)In Figure 2e, the authors observed n-LMR and this result was considered to be evidence of the chiral anomaly. Whether the authors perform the angular dependent MR measurements to check whether this n-LMR is sensitive to the field direction or not? In fact, the n-LMR may have different origins!

4)In Figure 2d and Figure 3c, the authors made the extrapolation from the Landau fan plots and obtained the evidences of the non-zero Berry phase. Based on these data, they made the conclusion of topological nature of the sample. To my knowledge, this conclusion is very weak because the index number is more than 20, any fit is very unreliable from Figure 2d (do the authors believe

these evidences firmly?). I suggest to delete the related statement based on these very weak evidences.

Reviewer #2 (Remarks to the Author):

The manuscript by E. Cheng et al. reports the magneto-transport and ARPES study of EnAs_3 , combined with the first-principles band structure calculations. From the magneto-transport measurements the authors have found non-zero Berry phase and negative longitudinal magnetoresistance in the antiferromagnetic state, whereas in the paramagnetic state, they found extremely large magnetoresistance. From the observation of electron and hole pockets around the Y point of bulk Brillouin zone, they have concluded inverted band structure and associated nodal-line structure originating from the band crossings of As p valence and conduction bands, consistent with their first-principles band-structure calculations. In my opinion, this manuscript is well written, data analyses are carefully done, and their data interpretation is reasonable. On the other hand, some of the authors' important claims do not seem to be well supported by their data. Thus, I have a reservation to recommend publication of this manuscript in Nature Communications in the current version. My specific comments are listed below:

1) Although the authors have suggested nonzero Berry phase by linearly extrapolating Landau index as a function of $1/B$ in Figs. 2d and 3c, it is unclear to what extent such extrapolation can be trusted. This is mainly because the small $1/B$ region is not covered in their plots. The authors should show error bars and explicitly state the actual experimental uncertainty on the value of intercepts in their plots. This point would be crucial to convince readers of the topological/non-topological nature of the observed energy bands.

2) ARPES data are too poor to support the existence of nodal line. I can see from Fig. 4b that there exists a 3D electron pocket centered at the Y point as well as the holelike band in wide k_y area of bulk Brillouin zone. However, it is unclear purely from the experimental data whether or not these bands indeed form the nodal line. It would be necessary to experimentally demonstrate the nodal line without the assistance of guidelines from the band calculations, by analyzing data more carefully the band crossing points. For example, the authors could analyze the peak positions in the MDCs and EDCs at several momentum cuts at different photon energies (and k_x) to convince readers of the essential existence of nodal line. I also expect that the bands should not degenerate at the momentum cuts with the non-zero k_x values (otherwise it would become nodal sphere).

3) Although the band calculations have predicted a drumhead surface state within the nodal loop bottomed at the binding energy of 0.15 eV, the authors were unable to clarify it. I suggest the authors to explain why they do not see the topological surface state.

Reviewer #3 (Remarks to the Author):

Reviewing report:

In the manuscript entitled “Magnetism-induced topological transition in EuAs₃”, the authors have successfully observed the topological protected states such as topological nodal-line semimetal state and topological massive Dirac state in the antiferromagnetic (AFM) EuAs₃ in experiments, and these nontrivial topologies are confirmed by the band calculations, Z₂ number calculations and Berry-phase synthesis from the theoretical point of view. We well know that to synthesize and to discover the magnetic topological materials is still a hard task both in the condensed-matter physics and in material science, and the present work is helpful to understand the topological quantum states in magnetic materials, and to develop the research field of magnetic topologies. Thus, I think that this work deserves to be accepted for publication in Nat. Commun.

However, before the formal acceptance, the following comments are suggested for authors to address or take into account:

(i) As for EuAs₃, the authors mentioned several different magnetic states, such as the AFM, the paramagnetic state, and the spin-polarized state, even the high-pressure state. So many states make the quantum states of the materials too complicated, I suggest that authors clarify the ground state and the excited states by comparing their lowest energy in different states obtained from the first-principles calculations, and make clear how to realize these states.

(ii) The authors claim that there is a TNL semimetal in paramagnetic or the spin-polarized state while a topological massive Dirac metal in AFM ground states. We know that the different topological nature in materials is determined by their structural symmetries. Thus for the EuAs₃ with different magnetic states, the symmetries and the band structures may be different, resulting in different topological features. Why do the authors determine that the transitions of these different topological states are induced by the magnetism, not by the high pressure? Since the TNL semimetal state exists in its high-pressure phase.

(iii) In the first section of the part of Results, the authors state that “...In magnetic systems, TRS is broken”, which means that the time-reversal symmetry is broken in magnetic systems. However, in its following sentence “... To preserve the Dirac node, extra symmetries, for example the combination of inversion (I) and time-reversal (T) symmetries, i.e., IT, are necessary ...”. Are these two statements contradictory? How to clarify this point?

(iv) Some statements or explanations seem unclear to readers. For example, in the last paragraph of the subsection “Topological properties in the AFM state”, the authors gave a description that “... while the data at lower temperatures don’t fit as well, which may be ascribed to the effect that magnetic transitions have on the chiral current ...”. How to determine that the changing on chiral currents by the magnetic transitions, not by the material structures? Or by the topological features?

(v) In the last paragraph of the subsection “Topological nodal-line structure in the paramagnetic state”, the authors described that “The verification of the nodal-line structure in the paramagnetic state serves as strong evidence for the existence of nodal-line structure in the spin-polarized state ...” I think this statement is doubtful. If we apply an external magnetic field to the paramagnetic materials to realize the spin-polarized states, the time-reversal symmetry will be destroyed in the later one, and thus, the nodal line will be broken in the spin-polarized state. How do the authors clarify this issue?

(vi) Some figures are suggested to revise or improve. For example, in the figures 1(f) and 1(g), the notices “WCC” should be close to the data axis, or be not in the same line of the figure number “f” or “g”, which may mislead readers.

(vii) Some sentences should be checked carefully. For example, in the first sentence of the subsection “Topological nodal-line structures in the spin-polarized state”, “..., where in Fig. 1(b) we have already observed a clear change in the quantum oscillations...”. However, in Fig. 1(b), I can’t find any information on quantum oscillations.

List of changes to manuscript

In the main text :

- (1) The article format has been modified.
- (2) Figure 4 and Supplementary Figure 5 have been updated, and captions have been also updated.
- (3) Page 6, the second paragraph:
“Therefore, the γ^1 and γ^2 bands...In order to validate the topological nature, more solid evidences are needed.” have been added.
- (4) Page 7, the last paragraph:
“The data above the ordering temperature... topological transitions.” have been updated.
- (5) Page 8, the last paragraph:
“Although the intercepts from the fit...and will require further verification.” have been added.
- (6) Page 10, the second paragraph:
Those sentences regarding to new ARPES results have been updated.
- (7) Page 11, the second paragraph:
“The verification of the nodal-line structure ...but the similarities may not end there.”
Have been changed to “The verification of the nodal-line structure...Nodal-line structure is thus strongly expected exist in spin-polarized EuAs_3 .”.
- (8) Reference section:

The references in the Supplementary Information have been moved to the main text.

In the Supplementary Information:

- (1) The article format has been modified.
- (2) The ARPES results in the main text in previous version have been moved to the Supplementary Information.

Detailed response to Referees' reports

Reviewer #1:

The manuscript reported the *ab initio* band calculations, electrical transport and angle-resolved photoemission spectroscopy (ARPES) measurements on the magnetic semimetal EuAs₃. The key point is that they demonstrated a magnetism-induced topological transition from a topological nodal-line semimetal in the paramagnetic or the spin-polarized state to a topological massive Dirac metal in the antiferromagnetic (AFM) ground state at low temperature. Also, a temperature-induced Lifshitz transition accompanied by the emergence of a new band below 3 K is revealed. All of these results are interesting. However, except for the evidence of the topological nodal-line semimetal in paramagnetic state can be recognized by ARPES, which is corroborated by band calculation, other assignments or statements actually still remain uncertainty.

Response: First of all, we appreciate Reviewer#1 for this evaluation of our results as interesting. We also carefully revised the manuscript to address his/her questions. To firmly support our conclusions, we also improved the quality of ARPES data as well as the *ab initio* calculations and carefully compared them.

1) The high field Shubnikov-de Haas (SdH) data at different temperatures sounds great, it presents very strong quantum oscillations in the spin polarized state. From Fig. 3a and 3b, the oscillation patterns at 15 K which is above the AFM transition T_N sound the same with those in the spin polarized state below T_N , I guess this is why the authors claim a magnetism-induced topological transition from a topological nodal-line semimetal in the spin-polarized state to a topological massive Dirac metal in the antiferromagnetic (AFM) ground state at low temperature. My concerning is that: is the band structure in this spin polarized state below T_N actually the same with that in the paramagnetic state above T_N (eg., the Nodal-line structures at the Y point in the Brillouin zone are proposed in both the spin-polarized and paramagnetic states)? In Figure 3h, the Hall data shows a hole-dominated state in paramagnetic state, but electron-dominated state in AFM or spin-polarized state below T_N .

Response: We thank Reviewer#1 for this valuable comment. To better show the FFT results below and above T_N , we redraw the plot, as shown in Fig. R1, and the data at 1.8 K, 15 K and 20 K are taken from Fig. 3(b). The FFT amplitudes at 15 K and 20 K are multiplied by a constant, respectively. As one can see, the FFT frequencies for ε

pocket are 346 ± 30 T, 330 ± 38 T, and 314 ± 44 T for 1.8 K, 15 K, and 20 K, respectively. The uncertainty is the full width at half maximum. These three FFT frequencies agree with each other in consideration of the uncertainty, and the difference may arise from the deduction of background and the oscillation intensity. Other frequencies for ξ , α' , η can be also recognized, although the frequency intensity at 15 K and 20 K is pretty weak and the uncertainty would also increase. Therefore, the band structure in the spin polarized state below T_N is consistent with that in the paramagnetic state above T_N .

Fig. R1 Comparison of FFT results at 1.8 K, 15 K and 20 K. Data are taken from Fig. 3(b).

However, for EuAs_3 in the AFM state, external magnetic field ~ 10.5 T could conquer the AFM coupling, and align the magnetic moments completely, driving AFM state to ferromagnetic state (i.e., the spin polarized state). For the paramagnetic state above T_N , external magnetic field can easily align the magnetic moments, and therefore drives the system into the spin polarized state. Given that, in either case (temperature below or above T_N), external magnetic field can drive the system into the spin polarized state, which is consistent with our results. Therefore, we can only obtain the information on Fermi surfaces and topology in the spin polarized state through SdH oscillations, while it needs to be studied in zero field by other techniques, for example ARPES.

2) In Figure 3a, the quantum oscillations are well resolved over the full magnetic field range in the measured temperatures, why the authors only treat the SdH data in the high-field range above B_m , while the low field data was selected from PPMS measurements. Maybe the low field data is noisy, do the authors try to make FFT from Figure 3a in the full field range? Do they find 8 different frequencies or only 4 frequencies?

Response: Thanks for your comment. Figure R2(a) shows the low-field quantum oscillations measured in a water-cool magnet (WM) and a superconducting magnet (SM). As one can see, the low-field data measured in a WM is too noisy. The noise is pronounced in the ΔR vs $1/B$ picture, as shown in Fig. R2(b). Regardless of the noise, we still analyzed the oscillatory components below B_m ($2.857 \text{ T} < B < 7 \text{ T}$), and the FFT frequency has been extracted, as displayed in Fig. R2(c). According to the data measured in a SM (Fig. 2(c) and Fig. R2(c)), three frequencies around 200 T have been recognized. For the data measured in a WM, in addition to the peaks stemming from noise, two small peaks around 200 T (in the red dotted box) can be observed, which gives the hint that the frequency with uncertainty consistent with that in Fig. 2(c) can be checked.

Fig. R2 **a** The magnetoresistance (MR) under 9 T measured in a water-cool magnet (WM). Data are taken from Fig. 3(a). The MR data measured in a superconducting magnet (SM) is also plotted, and data is taken from Fig. 2(b). The oscillatory components R_{xx} (**b**) and FFT results (**c**) below B_M for 1.8 K measured in a WM and 1.5 K measured in a SM. We used the data range from 2.857 T to 7 T to analyze the oscillatory components. **d** Comparison of the FFT results measured in a WM and a SM.

To check whether the FFT results above B_m measured in a SM are consistent with those measured in a WM, i.e., Fig. 3(b), we analyze the oscillatory components range from 11 T to 13 T (Fig. S1(c)), and the results are exhibited in Fig. R2(d). As one can see, the FFT frequency is totally different from that below B_m (i.e., in Fig. 2(c)), and two main peaks can be recognized at ~ 100 T and ~ 346 T, which are corresponding to ξ and ε pockets, respectively. Due to the limited period of oscillation, the peaks above

B_m measured in a SM are much broad. To check this, we also conducted the FFT analysis for the oscillatory components range from 11 T to 13 T measured at 1.8 K in a WM (Fig. R2(d)). As one can see, the result is consistent with that at 0.3 K, 1.5 K or 3 K. Therefore, we argue that the results measured in a SM are consistent with those measured in a WM. Moreover, we also report a temperature-induced Lifshitz transition below 3 K, as shown in Fig. S5(a). The frequency of the emerged pocket is 374 T, deduced from Hall oscillations. However, due to the broad peak at ~ 346 T, the new frequency below 3 K cannot be recognized from longitudinal oscillations.

3) In Figure 2e, the authors observed n-LMR and this result was considered to be evidence of the chiral anomaly. Whether the authors perform the angular dependent MR measurements to check whether this n-LMR is sensitive to the field direction or not? In fact, the n-LMR may have different origins!

Response: Thanks for your comment and reminding. Figure S2(c) shows the polar plots of the angular-resolved magnetoresistance (AMR) of EuAs₃ single crystal at 2 K. For $\varphi = 0^\circ$, the magnetic field is parallel to the [110] direction in the ab plane, i.e., perpendicular to the electric current I . For $\varphi = 90^\circ$, the magnetic field is parallel to the electric current I . At 90° , a small negative MR is observed. The magnitude of MR in this device is smaller than that in Fig. 2(e). The single crystals in Fig. 2(e) and Fig. S2(c) come from the same batch, so their qualities are nearly same. Therefore, the difference of negative MR may come from the deviation between the electric current I and magnetic field B , i.e., the I and B are not very parallel in a strict way. According to the results in Fig. 2(e) and Fig. S2(c), one can expect that the negative MR is sensitive to the field direction. So, we claim that the negative MR observed in EuAs₃ comes from chiral anomaly rather than magnetic scattering, and the reasons have been discussed in the main text.

4) In Figure 2d and Figure 3c, the authors made the extrapolation from the Landau fan plots and obtained the evidences of the non-zero Berry phase. Based on these data,

they made the conclusion of topological nature of the sample. To my knowledge, this conclusion is very weak because the index number is more than 20, any fit is very unreliable from Figure 2d (do the authors believe these evidences firmly?). I suggest to delete the related statement based on these very weak evidences.

Response: We appreciate Reviewer #1 and agree with this comment. The extrapolation from the Landau fan plots may have bigish uncertainty due to the large index number, although the intercepts with error bar fall in the range of $\pm 1/8$ (nontrivial Berry phase). In the AFM state, in addition to the nonzero Berry phase deduced from Landau fan plots, the negative MR could also serve as a smoking gun for topology. Therefore, in consideration of the band calculations, negative MR and “nonzero Berry phase”, the topological nature in the AFM ground state can be confirmed. In other words, the nonzero Berry phase is not the only evidence for topology in the AFM state. However, in the spin-polarized state, except for “nonzero Berry phase” (Fig. 3(c)), there is no direct evidence that we can provide to determine the topological nature, although four FFT frequencies from SdH are consistent with band calculations. For other magnetic topological semimetals, for example CeSb, the intercept deduced from large Landau index number (more than 20) is also used to determine the nontrivial Berry phase [1]. Therefore, the intercept can be still served as a strong hint for topology together with band calculations. Moreover, the unsaturated XMR is pretty unusual, which may arise from the nontrivial band topology (see the discussions in the main text). Besides, for topological nodal-line semimetal, the Berry phase is angle-dependent, as we observed in EuAs₃. Therefore, in combination with the DFT calculations and SdH results, we propose that EuAs₃ in the spin-polarized state is a topological nodal-line semimetal. And we hope more work to confirm the nodal-line structure in the future.

According to Reviewer #1's suggestion, we deleted the related statement and rewrote some sentences to soften a bit. The first paragraph on page 8, these sentences, “Therefore, the γ^1 and γ^2 bands may be topologically protected, while the other two are trivial. However, the index number is more than 20 (α , β and γ^1 pockets),

and hence the extrapolation from the Landau fan plots may have bigish uncertainty. In order to validate the topological nature, more solid evidences are needed.” have been revised and added. The second paragraph on page 11, these sentences, “Although the intercepts from the fit with large Landau index number cannot serve as a smoking gun for topology, we still use them. Because it is difficult for us to evaluate the topological nature in the fully-spin-polarized state above 11.1 T (the critical field is deduced from specific heat in Supplementary Fig. 1(c)). The intercepts give a hint that the ξ band is topologically protected, while the α' and η bands are topologically trivial. The intermediate value for ε is suggestive of a possible nontrivial Berry phase, but does not allow a strong conclusion, and will require further verification.” have been revised and added.

References

[1] Fang, Y. *et al.* Magnetic-field-induced nontrivial electronic state in the Kondo-lattice semimetal CeSb. *Phys. Rev. B* **101**, 094424 (2020).

Reviewer #2:

The manuscript by E. Cheng et al. reports the magneto-transport and ARPES study of $EnAs_3$, combined with the first-principles band structure calculations. From the magneto-transport measurements the authors have found non-zero Berry phase and negative longitudinal magnetoresistance in the antiferromagnetic state, whereas in the paramagnetic state, they found extremely large magnetoresistance. From the observation of electron and hole pockets around the Y point of bulk Brillouin zone, they have concluded inverted band structure and associated nodal-line structure originating from the band crossings of As p valence and conduction bands, consistent with their first-principles band-structure calculations. In my opinion, this manuscript is well written, data analyses are carefully done, and their data interpretation is reasonable. On the other hand, some of the authors' important claims do not seem to be well supported by their data. Thus, I have a reservation to recommend publication of this manuscript in Nature Communications in the current version. My specific comments are listed below:

Response: We thank Reviewer#2 for the valuable comments. We carefully considered

his/her concerns and accordingly significantly revised the manuscript to address these questions. We expect that the revisions could solve your concerns.

1) *Although the authors have suggested nonzero Berry phase by linearly extrapolating Landau index as a function of $1/B$ in Figs. 2d and 3c, it is unclear to what extent such extrapolation can be trusted. This is mainly because the small $1/B$ region is not covered in their plots. The authors should show error bars and explicitly state the actual experimental uncertainty on the value of intercepts in their plots. This point would be crucial to convince readers of the topological/non-topological nature of the observed energy bands.*

Response: We appreciate Reviewer #2 for this valuable comment. Reviewer #1 also raised the same concerns, see question #4 on page 6. The error bars from the fits have been already added in the previous version. In the first paragraph on page 8 in the main text, ***“To check their topological nature, a Landau index fan diagram is plotted in Fig. 2(d), yielding intercepts of 0.6(2), 0.5(1), -0.03(9), and 0.07(8) for α , β , γ^1 , and γ^2 , respectively.”*** In the first paragraph on page 11 in the main text, ***“To identify the topological nature of the four bands seen in quantum oscillations, a Landau index fan diagram is plotted in Fig. 3(c), and the intercepts are -0.0(1), 0.67(3), 0.34(4), and 0.61(4) for the ξ , α' , ε , and η bands, respectively.”*** For nonzero Berry phase, the intercepts fall in the range of $\pm 1/8$, while $3/8 \sim 5/8$ for trivial Berry phase. However, to avoid ambiguity, we deleted some related statements and rewrote some sentences to soften a bit.

2) *ARPES data are too poor to support the existence of nodal line. I can see from Fig. 4b that there exists a 3D electron pocket centered at the Y point as well as the holelike band in wide k_y area of bulk Brillouin zone. However, it is unclear purely from the experimental data whether or not these bands indeed form the nodal line. It would be necessary to experimentally demonstrate the nodal line without the assistance of guidelines from the band calculations, by analyzing data more carefully the band crossing points. For example, the authors could analyze the peak positions in the MDCs and EDCs at several momentum cuts at different photon energies (and k_x) to convince readers of the essential existence of nodal line. I also expect that the bands should not degenerate at the momentum cuts with the non-zero k_x values (otherwise it*

would become nodal sphere).

Response: Thanks for your comments and suggestions. We re-measured the electronic structure of EuAs₃ at 12 K, as shown in Fig. R3. We also updated Fig. 4 in the main text, and the results collected at 18 K have been moved into the Supplementary Information (Supplementary Fig. 5).

From the intensity plot of the Fermi surface at 12 K in the k_y - k_z plane (Fig. 4(a)) taken at E_F -0.2 eV, the pocket centered at the Y point (54 eV) can be easily identified, and two nodes arising from the crossing of the electron- and hole-like bands can be also observed in Fig. 4(b), which agrees with the band calculations. As observed in the analogue SrAs₃^[2], the drumhead-like surface state of EuAs₃ is buried in the bulk state, so it cannot be resolved by ARPES. For ARPES cut away from the Y point, the band-crossing area shrinks gradually and finally disappears, and the topological non-trivial nodal loop encircles the Y point, illustrated by the red ellipse in the Fig. 4(d). Besides, the k_y -dependent evolution of the band structure shows a good agreement with the calculations (the black dotted curves in Fig. 4(d)) and the corresponding energy-distribution curves (EDCs) could further confirm the nodes introduced by the band crossing and their k_y -dependent evolution in Fig. 4(f) (dashed line is a guide for the eyes to trace the dispersions). The evolution of the nodes along the k_x -direction is presented in Fig. 4(e), which shows the band dispersions along cuts 1-4 indicated in Fig. 4 (b) (photoemission intensity map of constant energy contours at 0.5eV below E_F). We noticed both the electron- and hole-like bands deplete their spectral weight from 1-4, consistent with the band-crossing scenario. However, whether or not a gap opens in the nodes away from $k_x = 0$ remains vague due to the intrinsic broadness of the electronlike band. We also measured the electronic structure of another sample at 18 K, as shown in Supplementary Fig. 5, and obtain the same results. These results have been updated on Page 10 in the main text.

Fig. R3 The topological non-trivial nodal loop visualized by ARPES. a Photon energy dependent plot of photoemission intensity in the k_z - k_y plane taken at $E_F - 0.2$ eV. **b** Photoemission intensity map of constant energy contours at 0.5 eV below E_F in the k_x - k_z plane, the data was collected using photons with $h\nu = 55$ eV. **c** The Brillouin Zone (BZ) of EuAs_3 , with high-symmetry points and (010) surface labeled. **d** The band dispersions along k_y direction probed by different photon energies. The calculations plotted by the black dotted curves superimposed on the experimental electronic structure. The red ellipse illustrates the topological non-trivial nodal loop schematically. **e** The band dispersions along cuts 1-4 as indicated in (b), respectively. **f** corresponding energy-distribution curves (EDCs) taken at different photon energies.

3) Although the band calculations have predicted a drumhead surface state within the nodal loop bottomed at the binding energy of 0.15 eV, the authors were unable to clarify it. I suggest the authors to explain why they do not see the topological surface state.

Response: Thanks for the comment. According to the DFT calculations, the electronic structure of EuAs_3 highly resembles that of SrAs_3 , especially near E_F ^[2]. For SrAs_3 , the surface state is invisible since it is buried in the bulk states^[2]. Therefore, ARPES cannot observe the topological surface state in SrAs_3 ^[2]. For EuAs_3 , like the case in SrAs_3 , the surface state is also buried in the bulk states, which cannot be resolved by our ARPES

experiments.

References

[2] Song, Y. K. *et al.* Photoemission spectroscopic evidence for the Dirac nodal line in monoclinic semimetal SrAs₃. *Phys. Rev. Lett.* **124**, 056402 (2020).

Reviewer #3:

In the manuscript entitled "Magnetism-induced topological transition in EuAs₃", the authors have successfully observed the topological protected states such as topological nodal-line semimetal state and topological massive Dirac state in the antiferromagnetic (AFM) EuAs₃ in experiments, and these nontrivial topologies are confirmed by the band calculations, Z₂ number calculations and Berry-phase synthesis from the theoretical point of view. We well know that to synthesize and to discover the magnetic topological materials is still a hard task both in the condensed-matter physics and in material science, and the present work is helpful to understand the topological quantum states in magnetic materials, and to develop the research field of magnetic topologies. Thus, I think that this work deserves to be accepted for publication in Nat. Commun. However, before the formal acceptance, the following comments are suggested for authors to address or take into account:

Response: Thanks for your positive comments on our work. We carefully considered your suggestions and concerns for further improvement the manuscript and accordingly revised the manuscript. We expect that the revisions could meet your requirement.

1) As for EuAs₃, the authors mentioned several different magnetic states, such as the AFM, the paramagnetic state, and the spin-polarized state, even the high-pressure state. So many states make the quantum states of the materials too complicated, I suggest that authors clarify the ground state and the excited states by comparing their lowest energy in different states obtained from the first-principles calculations, and make clear how to realize these states.

Response: We thank Reviewer #3 for this valuable suggestion.

Reviewer #3 suggests us to clarify the ground state and the excited states of EuAs_3 by comparing their lowest energy in different states obtained from the first-principles calculations. According to his/her suggestions, we performed energy calculations for various magnetic configurations including one ferromagnetic (FM) order and six antiferromagnetic (AFM) configurations, as shown in Fig. R4. The resulted magnetic energies are given in Table S1. Our first-principles calculations predict that the AFM1 configuration has the lowest total energy. This agrees with the magnetic structure observed in earlier neutron diffraction experiments on EuAs_3 ^[3]. Therefore, the AFM1 phase is the magnetic ground state of EuAs_3 .

Antiferromagnetically ordered EuAs_3 can undergoes a phase transition to a spin-polarized state upon the application of an external magnetic field, because the magnetic moments of all unbound electrons will eventually line up with the applied magnetic field, if the external magnetic field is large enough, as discussed in the main text. The paramagnetic state can be realized in EuAs_3 by increasing the temperature higher than the magnetic transition temperature of EuAs_3 .

Fig. R4. The six antiferromagnetic configurations considered for EuAs_3 . The blue and magenta balls represent Eu atoms with opposite moment directions which are parallel to the b axis.

Config.	Energy (eV)
FM	-26.90974968
AFM1	-26.91385426
AFM2	-26.90928858
AFM3	-26.91061771
AFM4	-26.91329305
AFM5	-26.91000387
AFM6	-26.90834134

Table S1. Total energies (in unit of eV/f.u.) of seven different magnetic structures for EuAs₃ calculated by GGA+*U*+SOC method.

References

[3] Chattopadhyay, T., Schnering, H. G. v. and Brown, P. J. Neutron diffraction study of the magnetic ordering in EuAs₃. *J. Magn. Magn. Mater.* **28**, 247-249 (1982).

2) *The authors claim that there is a TNL semimetal in paramagnetic or the spin-polarized state while a topological massive Dirac metal in AFM ground states. We know that the different topological nature in materials is determined by their structural symmetries. Thus for the EuAs₃ with different magnetic states, the symmetries and the band structures may be different, resulting in different topological features. Why do the authors determine that the transitions of these different topological states are induced by the magnetism, not by the high pressure? Since the TNL semimetal state exists in its high-pressure phase.*

Response: We appreciate Reviewer #3 for this valuable comment. In this work, we explore the magnetic effect rather than pressure effect. For topological semimetals, pressure can suppress XMR and induce some exotic properties. For example, in SrAs₃, we observed a pressure-induced Lifshitz transition, structural phase transition and superconductivity^[4]. For the high-pressure phase of SrAs₃, band calculations proposed that it is a topological crystalline insulator, featuring several criteria^[4]. For a magnetic

system, pressure could change the magnetic configuration. Previous neutron-diffraction studies on EuAs_3 single crystal uncovered a pressure-induced spiral phase with magnetic moments modulated in the (010) plane, and two triple points in the P - T phase diagram^[5,6]. The magnetic configuration of EuAs_3 under pressure is very complicated, and it would increase the difficulty of calculations. We hope our work could inspire people to explore the high-pressure properties of EuAs_3 .

References

[4] Cheng, E. J. *et al.* Pressure-induced superconductivity and topological phase transitions in the topological nodal-line semimetal SrAs_3 . *npj Quantum Mater.* **5**, 38 (2020).

[5] Chattopadhyay T., Brown P. J. New high-pressure spiral phase of EuAs_3 . *Phys. Rev. B* **36**, 2454 (1987).

[6] Chattopadhyay T., Brown P. J. Neutron-diffraction study of the pressure-temperature phase diagram of EuAs_3 . *Phys. Rev. B* **41**, 4358 (1990).

3) *In the first section of the part of Results, the authors state that "...In magnetic systems, TRS is broken", which means that the time-reversal symmetry is broken in magnetic systems. However, in its following sentence "... To preserve the Dirac node, extra symmetries, for example the combination of inversion (I) and time-reversal (T) symmetries, i.e., IT, are necessary ...". Are these two statements contradictory? How to clarify this point?*

Response: We thank Reviewer #3 for this valuable comment. These two statements are not contradictory because the combined effective IT symmetry can be preserved, although the I symmetry and the T symmetry are each broken^{7,8}. In Dirac materials, two doubly degenerate bands contact at discrete momentum points called Dirac points, and disperse linearly along all directions around these points. The four-fold degenerate Dirac points are unstable by themselves; hence, symmetry protection is necessary. When I or T symmetry, or both, is broken and the double-band degeneracy is lifted, the touching points of two non-degenerate bands can form a 3D Weyl semimetal rather

than a 3D Dirac semimetal. Zhang and his colleagues⁷ presented the conceptually intriguing proposal that Dirac fermions can also be hosted even in antiferromagnetic materials, where both I and T are broken but their combination IT is preserved. The combined effective IT symmetry ensures each band is doubly degenerate, giving rise to the possibility of the formation of four-fold degenerate Dirac points.

References

[7] Tang, P. Zhou, Q. Xu, G. and Zhang, S.-C. Dirac fermions in an antiferromagnetic semimetal, *Nat. Phys.* **12**, 1100 (2016).

[8] Šmejkal, L. Železný, J. Sinova, J. and Jungwirth, T. Electric Control of Dirac Quasiparticles by Spin-Orbit Torque in an Antiferromagnet. *Phys. Rev. Lett.* **118**, 106402 (2017).

4) Some statements or explanations seem unclear to readers. For example, in the last paragraph of the subsection "Topological properties in the AFM state", the authors gave a description that "... while the data at lower temperatures don't fit as well, which may be ascribed to the effect that magnetic transitions have on the chiral current ...". How to determine that the changing on chiral currents by the magnetic transitions, not by the material structures? Or by the topological features?

Response: We thank Reviewer #3 for this very good comment. For SrAs₃, the nonmagnetic analog to EuAs₃, C_w decreases monotonically^[9], as we observed in EuAs₃, when $T > T_N$. However, for EuAs₃, C_w displays a dip around T_N , which hasn't been reported in SrAs₃. In other word, the possibility that the anomaly in C_w comes from material structures can be excluded. In the text, we proposed that the anomaly arises from magnetic transitions. However, we cannot exclude the possibility that the anomaly stems from topological features. We fit the data below 3 T to get the C_w . However, when external magnetic field is applied in ab plane, the magnetic configuration varies significantly^[10], as observed in Fig. S5. And we cannot exclude how and to what extent the topological features change in these intermediate states, which is hard for us to

determine. Therefore, the anomaly in C_w may come from magnetic transitions or topological transitions.

In the main text, we change the sentence “*The data above the ordering temperature are well described by the ABJ equation, while the data at lower temperatures don’t fit as well, which may be ascribed to the effect that magnetic transitions have on the chiral current.*” to “*The data above the ordering temperature are well described by the ABJ equation, while the data at lower temperatures don’t fit as well, which may be ascribed to magnetic transitions or topological transitions.*”

Fig. R5. Magnetic phase diagram of EuAs_3 [10]. The magnetic field is applied along b axis.

References

- [9] Li, S. C. *et al.* Evidence for a Dirac nodal-line semimetal in SrAs_3 . *Sci. Bull.* **63**, 535-541 (2018).
- [10] Chattopadhyay, T. and Brown, P. J. Field-induced transverse-sine-wave-to-longitudinal- sine-wave transition in EuAs_3 . *Phys. Rev. B* **38**, 795 (1988).

5) In the last paragraph of the subsection “Topological nodal-line structure in the paramagnetic state”, the authors described that “The verification of the nodal-line structure in the paramagnetic state serves as strong evidence for the existence of

nodal-line structure in the spin-polarized state ...” I think this statement is doubtful. If we apply an external magnetic field to the paramagnetic materials to realize the spin-polarized states, the time-reversal symmetry will be destroyed in the later one, and thus, the nodal line will be broken in the spin-polarized state. How do the authors clarify this issue?

Response: We thank Reviewer #3 for this valuable comment. We fully agree with the reviewer’s opinion. To avoid this doubtful statement, we change the sentence “***The verification of the nodal-line structure in the paramagnetic state serves as strong evidence for the existence of nodal-line structure in the spin-polarized state, which has closely similar but spin-split band structure, but the similarities may not end there.***” to “***The verification of the nodal-line structure in the paramagnetic state by utilizing ARPES measurements and DFT calculations shows remarkable agreement between the theoretical and experimental values. For the spin-polarized state, which is predicted to hosts closely similar but spin-split band structure to the paramagnetic state. Nodal-line structure is thus strongly expected to exist in spin-polarized EuAs₃.***” on Page 11.

6) Some figures are suggested to revise or improve. For example, in the figures 1(f) and 1(g), the notices “WCC” should be close to the data axis, or be not in the same line of the figure number “f” or “g”, which may mislead readers.

Response: We thank Reviewer #3 for this comment. The Figs. 1(f) and 1(g) have been updated.

7) Some sentences should be checked carefully. For example, in the first sentence of the subsection “Topological nodal-line structures in the spin-polarized state”, “..., where in Fig. 1(b) we have already observed a clear change in the quantum oscillations...”. However, in Fig. 1(b), I can’t find any information on quantum oscillations.

Response: We thank Reviewer #3 for pointing out this typo. “Fig. 1(b)” should be “Fig. 2(b)”, and we have fixed the error. Thanks!

REVIEWER COMMENTS

Reviewer #1 (Remarks to the Author):

In this revised version, the authors answered most of my questions and I accept their suggestion that a magnetism-induced topological transition from a topological nodal-line semimetal in the paramagnetic or the spin-polarized state to a topological massive Dirac metal in the antiferromagnetic (AFM) ground state. However, there still have a few questions which needs to be clarified.

1)The authors claim a temperature-induced Lifshitz transition accompanied by the emergence of a new band below 3 K. Indeed, the Hall data in Fig.3h shows a clear slope change from positive to negative near 3 K. However, why the resistivity does not show any changes near 3 K? In contrast, the resistivity shows significant changes near the AFM transition temperature T_N , but the slope of the Hall resistivity does not show significant changes. These data sound contradicted! The current data does not provide any solid evidences for this T-induced LF transition.

2)If we agree the T-induced LF transition near 3K, why the MR oscillations in low field range do not show variations across 3K when the transport property switches from electron-dominated to Hall-dominated?

3)Does the AFM transition near T_N really strongly correlate to the massive Dirac electronic state below T_N ?

4)In page 6, the table I is very confusing if putting the 8-bands data for different field ranges in one table. I suggest to split the table to two or clearly ascribe the field range for the two groups.

Reviewer #2 (Remarks to the Author):

I think that the authors have reasonably answered my questions regarding the estimation of nonzero Berry phase and existence of nodal line. In particular, new data supplied by the authors show more clearly the existence of nodal loop. I also understand the authors' response that it is hard for ARPES to resolve the topological drumhead surface state because it is buried in the bulk band. In my opinion, the manuscript is appropriately revised. Now I would recommend the publication of this manuscript in Nature Communications in its present form.

Reviewer #4 (Remarks to the Author):

Since the authors have given their suitable responses to my all comments, and the manuscript and the corresponding supplemental material have been revised carefully and improve largely, the current version can be accepted for publication in Nat. commun.

List of changes to manuscript

In the main text:

(1) Table I has been modified, and magnetic field range for analysis has been added.

(2) According to the comments raised by Reviewer#1, we thereby revised some sentences regarding to the temperature-induced Lifshitz transition.

a) Page 11, the first paragraph:

“However, we also find evidence for an additional Lifshitz transition within the antiferromagnetic phase.” has been revised to “However, a possible additional Lifshitz transition below 3 K has also been suggested.”.

b) Page 12, the second paragraph:

“To check this, we analyze the oscillatory component ($\Delta\rho_{xy}$), and identify a new oscillation frequency of 374 T (denoted as the φ band) with trivial topology (see Supplementary Fig. 7(b)), demonstrating that a Lifshitz transition does indeed occur¹⁸. Temperature-induced Lifshitz transitions are also observed in other TSMs, for example, $M\text{Te}_5$ ($M = \text{Zr}, \text{Hf}$)^{27,28}, which have been used to explain the origin of the resistivity anomaly. However, no such anomaly can be observed in EuAs_3 , indicative of its unusual origin.” have been revised to

“Temperature-induced Lifshitz transitions are also observed in other TSMs, for example, $M\text{Te}_5$ ($M = \text{Zr}, \text{Hf}$)^{27,28}, $\text{InTe}_{1-\delta}$ ²⁹, ZrSiSe ³⁰, WTe_2 ³¹, and TaIrTe_4 ³². Anomalies in both longitudinal resistivity and Hall resistivity/coefficient can be found in $M\text{Te}_5$ ($M = \text{Zr}, \text{Hf}$)^{27,28}, $\text{InTe}_{1-\delta}$ ²⁹ and ZrSiSe ³⁰, but not in WTe_2 ³¹ and TaIrTe_4 ³². Since we have observed the change of Hall resistivity in EuAs_3 (Fig. 3(h)), we wonder how the longitudinal resistivity evolves with decreasing temperature. The inset of Fig. 2(a) shows the low-temperature resistivity from 0.3 to 2.5 K in zero field, and we did not observe any distinct anomaly. Considering that the variation in resistivity may be very weak, and the temperature range from 0.3 to 2.5 K is not appropriate, we measured two more samples (denoted as Sample 4 and Sample 5 in Supplementary Fig. 7(a)), and found that there is a very weak anomaly at ~ 2.3

K in resistivity for both samples (see Supplementary Fig. 7(b)). Since the Lifshitz transition should also manifest in quantum oscillations, we further check the low-field MR data below B_M in Fig. 2(b) and the FFT in Fig. 2(c). One can see that due to the limited oscillatory periods and/or noise at 3 K, the Lifshitz transition cannot be resolved from the low-field quantum oscillation data. To verify it, other low-temperature probes are needed, for example ARPES, STM/STS.

Now, we turn to the high-field state above B_M . The temperature dependence of Hall coefficient measured at 9 T for Sample 5 is shown in Supplementary Fig. 7(c). With decreasing temperature, a small peak at ~ 3.6 K arises and Hall coefficient changes its sign from positive to negative at ~ 2.3 K. We then analyze the oscillatory component ($\Delta\rho_{xy}$) above B_M , and a new oscillation frequency of 374 T (denoted as the φ band) can be clearly distinguished at 0.3 K, as shown in Supplementary Fig. 8(a). The trivial topology nature for φ band has also been demonstrated (see Supplementary Fig. 8(b)). Therefore, a temperature-induced Lifshitz transition likely exists in both antiferromagnetic and spin-polarized states, although the change of Fermi surface topology with temperature in these two states may be different.”.

c) Page 17, the References section:

Four references, “29. Back, S. Y. *et al.* Temperature-induced Lifshitz transition and charge density wave in $\text{InTe}_{1-\delta}$ thermoelectric materials. *ACS Appl. Energy Mater.* **3**, 3628-3636 (2020).

30. Chen, F. C. *et al.* Temperature-induced Lifshitz transition and possible excitonic instability in ZrSiSe . *Phys. Rev. Lett.* **124**, 236601 (2021).

31. Wu, Y. *et al.* Temperature-induced Lifshitz transition in WTe_2 . *Phys. Rev. Lett.* **115**, 166602 (2015).

32. Jian, Y. *et al.* Transport signatures of temperature-induced chemical potential shift and Lifshitz transition in layered type-II Weyl semimetal TaIrTe_4 . *2D Mater.* **8**, 015020 (2020).” have been added.

In the Supplementary Information:

(1) New Supplementary Note 7 has been added. Previous Supplementary Note 7 and Supplementary Note 8 are Supplementary Note 8 and Supplementary Note 9 in the revised manuscript

Detailed response to Referees' reports

Reviewer #1 (Remarks to the Author):

In this revised version, the authors answered most of my questions and I accept their suggestion that a magnetism-induced topological transition from a topological nodal-line semimetal in the paramagnetic or the spin-polarized state to a topological massive Dirac metal in the antiferromagnetic (AFM) ground state. However, there still have a few questions which needs to be clarified.

Response: First of all, we appreciate Reviewer #1 for this evaluation of our revised manuscript. We carefully revised the manuscript further to address his/her questions.

1) The authors claim a temperature-induced Lifshitz transition accompanied by the emergence of a new band below 3 K. Indeed, the Hall data in Fig. 3h shows a clear slope change from positive to negative near 3 K. However, why the resistivity does not show any changes near 3 K? In contrast, the resistivity shows significant changes near the AFM transition temperature T_N , but the slope of the Hall resistivity does not show significant changes. These data sound contradicted! The current data does not provide any solid evidences for this T -induced LF transition.

Response:

We thank Reviewer #1 for this valuable comment. Lifshitz transitions driven by pressure, chemical doping/substitution or strain are common, while only several cases that Lifshitz transition is driven by temperature have been suggested, for example in $M\text{Te}_5$ ($M = \text{Zr}, \text{Hf}$)

¹⁻³, InTe_{1-δ}⁴, ZrSiSe⁵, WTe₂⁶ and TaIrTe₄⁷, which can be explained by the temperature-induced chemical potential shift. Anomalies in both longitudinal resistivity and Hall resistivity/coefficient can be found in *M*Te₅ (*M* = Zr, Hf)¹⁻³, InTe_{1-δ}⁴, and ZrSiSe⁵, but not in WTe₂⁶ and TaIrTe₄⁷. For WTe₂, a temperature-induced Lifshitz transition associated with the complete disappearance of the hole pockets at ~160 K has been demonstrated⁶. However, the slope of the transverse Hall resistivity of WTe₂ remains negative from 300 K to 1.8 K (see Fig. R1(b)), and there is no clear anomaly in both Hall coefficient and longitudinal resistivity (Fig. R1(a)) at ~160 K. For TaIrTe₄, a temperature-induced Lifshitz transition has also been suggested⁷. As shown in Fig. R2(b), the slope of Hall resistivity changes its sign from positive to negative at ~30 K. However, no clear anomaly is observed in resistivity (Fig. R2(a)). Therefore, Lifshitz transition does not necessarily cause a clear anomaly on longitudinal resistivity.

Fig. R1 | Physical properties of WTe₂. **a** Resistivity data and magnetoresistance of WTe₂. **b** Hall coefficient of WTe₂ at 9 T. Inset shows Hall resistivity. Data are taken from Ref. [6].

Fig. R2 | Physical properties of TaIrTe₄. **a** Resistivity data of TaIrTe₄. **b** Hall resistivity.

Data are taken from Ref. [7].

Fig. R3 | Longitudinal and transverse resistivity of EuAs₃ single crystals. **a** The low-temperature longitudinal resistivity of two EuAs₃ single crystals. Two vertical arrows represent the temperatures where incommensurate antiferromagnetic (T_N) and incommensurate-to-commensurate lock-in (T_L) transitions take place. **b** The low-temperature longitudinal resistivity of these two single crystals measured in a ³He cryostat. In order to avoid the influence of noise, we fit the data by using a polynomial to reproduce the experimental data (the red solid lines). The olive and black lines are the derivative of the simulated data (the red solid lines) for Sample 4 and Sample 5, respectively. **c** The transverse Hall resistivity of EuAs₃ single crystal (Sample 5) at 9 T. The solid line represents the derivative of the data. Magnetic field is applied along [110] direction.

For EuAs_3 , the Hall resistivity changes its sign from positive to negative below 3 K. However, in the inset of Fig. 2(a) in the main text, we did not observe any distinct anomaly in resistivity below 2.5 K. Considering that the temperature range may be not appropriate, we measure two more samples (Sample 4 and Sample 5, the same batch as those used in the main text) to further check it. Figure R3(a) shows the low-temperature resistivity of them. The antiferromagnetic and incommensurate-to-commensurate lock-in transition temperatures are 11 K and 10.3 K, respectively. The low-temperature resistivity measurements from 0.3 to 5 K for Sample 4 and Sample 5 have been performed, as shown in Fig. R3(b), and there is no any distinct anomaly below 3 K. Considering that the variation in resistivity may be very weak, we calculate the derivative of the experimental data for Sample 4 and Sample 5, respectively. As displayed in Fig. R4, due to the noise, anomaly cannot be distinguished. To solve this problem, we use a polynomial to reproduce the experimental data first (the red solid lines in Fig. R3(b)), and then the derivatives for Sample 4 and Sample 5 have been deduced. A broad peak locating at ~ 2.3 K shows up for both samples. The peak in derivative suggests the change of slope in resistivity around 2.3 K. Thus, in addition to the Hall data, resistivity shows a very weak change below 3 K.

Fig. R4 | The derivative of the experimental data (Fig. R3(c)) for Sample 4 and Sample 5, respectively.

In the low field range (AFM state), the Lifshitz transition below 3 K cannot be resolved from the low-field quantum oscillation data due to limited oscillatory periods and noise, as shown in Fig. 2(b) and 2(c). However, Hall data and anomaly in resistivity below 3 K give a hint that temperature-induced Lifshitz transition takes place in the antiferromagnetic state. Under high magnetic field above B_M , the antiferromagnetic ordering will be suppressed, and a spin-polarized state is induced. The temperature dependence of Hall coefficient measured at 9 T for Sample 5 is shown in Fig. R3(c). With decreasing temperature, a small peak at ~ 3.6 K arises and Hall coefficient changes its sign from positive to negative at ~ 2.3 K. We then analyze the oscillatory component ($\Delta\rho_{xy}$) above B_M , and a new oscillation frequency of 374 T (denoted as the φ band) can be clearly distinguished at 0.3 K, as shown in Supplementary Fig. 8(a). Therefore, a temperature-induced Lifshitz transition likely exists in both antiferromagnetic and spin-polarized states, although the change of Fermi surface topology with temperature in these two states may be different.

The Fermi surfaces in the AFM state and the paramagnetic state are different, as shown in Fig. 1(d) in the main text and Supplementary Fig. 4(c) in the Supplementary Information, respectively. As discussed above, for WTe_2 [6], Lifshitz transition does not necessarily cause a significant change of the slope of Hall resistivity around 160 K (Fig. R1(b)). In this context, the change of Fermi surface around AFM transition in EuAs_3 does not necessarily cause a significant change of the slope of Hall resistivity too. As pointed out by Reviewer #1, the longitudinal resistivity of EuAs_3 around AFM transition shows significant change. This may be due to the decrease of magnetic scattering below the AFM transition.

We agree with Reviewer #1 that the temperature-induced Lifshitz transition in EuAs_3 needs to be verified further. And we hope our work will inspire more experiments in the future, for example low-temperature ARPES, STM/STS etc.

References

1. Zhang, Y. *et al.* Electronic evidence of temperature-induced Lifshitz transition and topological nature in ZrTe_5 . *Nat. Commun.* **8**, 15512 (2017).
2. Chi, H. *et al.*, Lifshitz transition mediated electronic transport anomaly in bulk ZrTe_5 . *New J. Phys.* **19**, 015005 (2017).

3. Zhang, Y. *et al.* Temperature-induced Lifshitz transition in topological insulator candidate HfTe_5 . *Sci. Bull.* **62**, 950-956 (2017).
4. Back, S. Y. *et al.*, Temperature-induced Lifshitz transition and charge density wave in $\text{InTe}_{1-\delta}$ thermoelectric materials. *ACS Appl. Energy Mater.* **3**, 3628-3636 (2020).
5. Chen, F. C. *et al.*, Temperature-induced Lifshitz transition and possible excitonic instability in ZrSiSe . *Phys. Rev. Lett.* **124**, 236601 (2021).
6. Wu, Y. *et al.* Temperature-induced Lifshitz transition in WTe_2 . *Phys. Rev. Lett.* **115**, 166602 (2015).
7. Jian, Y. *et al.*, Transport signatures of temperature-induced chemical potential shift and Lifshitz transition in layered type-II Weyl semimetal TaIrTe_4 . *2D Mater.* **8**, 015020 (2020).

2) If we agree the T-induced LF transition near 3 K, why the MR oscillations in low field range do not show variations across 3 K when the transport property switches from electron-dominated to hole-dominated?

Response:

We thank Reviewer #1 for this valuable comment. To better answer this question from Reviewer #1, we reproduce the data of SdH oscillations in low field range in Fig. R5. The data for Figs. R5(a), R5(b) and R5(c) are taken from Figs. 2(b), 2(c) and 3(h), respectively.

As shown in Fig. R5(a), with increasing temperature from 0.3 K to 3 K, the SdH oscillations are gradually weakened. The oscillations are quite weak at 3 K and above, so that we cannot tell whether there is a variation across 3 K or not. The FFT frequencies in the antiferromagnetic state cannot be clearly extracted at 3 K (Fig. R5(b)). We also check the Hall resistivity oscillations in low field range, as shown in Fig. R5(c), and analyze the SdH oscillations below B_M at 0.3 K and 1.5 K (Fig. R5(d)). Again, the oscillations are quite weak at 3 K and above. Due to the limited oscillatory periods and noise, the FFT frequencies in antiferromagnetic state cannot be distinguished for both 0.3 K and 1.5 K. Therefore, higher magnetic field is needed to obtain more oscillatory periods. Only in this way can the evolution of Fermi surface across 3 K be determined. However, higher

magnetic field would drive the antiferromagnetic state into spin-polarized state, and the Fermi surface will change accordingly. Therefore, other low-temperature measurements, such as ARPES and STM/STS, are needed to verify the temperature-induced Lifshitz transition with decreasing temperature.

Fig. R5 | Longitudinal and transverse resistivity and FFT results of EuAs_3 single crystals in low field range at various temperatures. **a** Magnetoresistance (MR) accompanied by distinct SdH oscillations. B_M represents the critical magnetic field which induces a magnetic transition from a collinear antiferromagnetic phase to a polarized ferromagnetic phase. **b** FFT results at various temperatures. The inset displays the oscillatory component ρ_{xx} below B_M . **c** Hall resistivity in low field range at various temperatures. **d** FFT results at 0.3 K and 1.5 K. The inset displays the oscillatory component ρ_{xy} below B_M . Data for **a**, **b** and **c** are taken from Figs. 2(b), 2(c) and 3(h), respectively.

3) Does the AFM transition near T_N really strongly correlate to the massive Dirac electronic state below T_N ?

Response:

We thank Reviewer #1 for this valuable comment. According to the DFT calculations, we proposed that EuAs₃ is a massive Dirac metal in the AFM state, while it is a topological node-line semimetal in the paramagnetic state or the spin-polarized state. Based on the SdH measurements, the Fermi surface in the AFM state is different from that in the spin-polarized state (Table I), indicating that the change of magnetic configuration induced by external field has a great effect on the Fermi surface as well as the topology. And the topological node-line semimetal state in the paramagnetic state has been demonstrated by our ARPES measurements. Therefore, we argue that AFM transition correlates to the massive Dirac electronic state below T_N . How and to what extent it affects the electronic structure and topology remains to be explored.

4) In page 6, the table I is very confusing if putting the 8-bands data for different field ranges in one table. I suggest to split the table to two or clearly ascribe the field range for the two groups.

Response: We thank Reviewer #1 for this valuable suggestion. Table I has been modified in the revised manuscript as suggested.

Reviewer #2 (Remarks to the Author):

I think that the authors have reasonably answered my questions regarding the estimation of nonzero Berry phase and existence of nodal line. In particular, new data supplied by the authors show more clearly the existence of nodal loop. I also understand the authors' response that it is hard for ARPES to resolve the topological drumhead surface state because it is buried in the bulk band. In my opinion, the manuscript is appropriately revised. Now I would recommend the publication of this manuscript in Nature Communications in its present form.

Response: We appreciate Reviewer #2 for his/her recommendation.

Reviewer #4 (Remarks to the Author):

Since the authors have given their suitable responses to my all comments, and the manuscript and the corresponding supplemental material have been revised carefully and improve largely, the current version can be accepted for publication in Nat. Commun.

Response: We thank Reviewer #4 for his/her positive evaluation and recommendation.

REVIEWERS' COMMENTS

Reviewer #1 (Remarks to the Author):

The authors have answered all of my questions. I have no more concerns for this manuscript, it is ready to acceptance for publication.